# Ephrin-B3 controls excitatory synapse density through cell-cell competition for EphBs

**Nathan T Henderson[1,2†], Sylvain J Le Marchand[3†], Martin Hruska[1], Simon Hippenmeyer[4], Liqun Luo[5], Matthew B Dalva[1]\***

[1]Department of Neuroscience, The Vickie and Jack Farber Institute for Neuroscience, Thomas Jefferson University, Philadelphia, United States; [2]Department of Neuroscience, University of Pennsylvania, Philadelphia, United States; [3]Bio-Imaging Center, University of Delaware, Newark, United States; [4]Institute of Science and Technology Austria, Klosterneuburg, Austria; [5]Department of Biology, Howard Hughes Medical Institute, Stanford University, Stanford, United States

**\*For correspondence:**
Matthew.Dalva@jefferson.edu

[†]These authors contributed equally to this work

**Competing interests:** The authors declare that no competing interests exist.

**Abstract** Cortical networks are characterized by sparse connectivity, with synapses found at only a subset of axo-dendritic contacts. Yet within these networks, neurons can exhibit high connection probabilities, suggesting that cell-intrinsic factors, not proximity, determine connectivity. Here, we identify ephrin-B3 (eB3) as a factor that determines synapse density by mediating a cell-cell competition that requires ephrin-B-EphB signaling. In a microisland culture system designed to isolate cell-cell competition, we find that eB3 determines winning and losing neurons in a contest for synapses. In a Mosaic Analysis with Double Markers (MADM) genetic mouse model system in vivo the relative levels of eB3 control spine density in layer 5 and 6 neurons. MADM cortical neurons in vitro reveal that eB3 controls synapse density independently of action potential-driven activity. Our findings illustrate a new class of competitive mechanism mediated by trans-synaptic organizing proteins which control the number of synapses neurons receive relative to neighboring neurons.
DOI: https://doi.org/10.7554/eLife.41563.001

## Introduction

In the developing brain, neurons must form specific connections in an environment with many potential synaptic partners while also controlling the number of synapses they receive (*Sanes and Yamagata, 2009*). Defects in the abundance of synaptic connections are associated with devastating diseases, such as schizophrenia, intellectual disability, and autism (*Yin et al., 2012*; *Zoghbi and Bear, 2012*). The precise number and pattern of contacts only emerge after a period of exuberant synaptogenesis, which generates an excess of synapses that are then subjected to further refinement (*Williams et al., 2010*). However, while synaptic sites are over-produced during development, connectivity is thought to remain sparse, with only a small number of all possible synaptic contacts made (*Druckmann et al., 2014*; *Song et al., 2005*; *Stepanyants and Chklovskii, 2005*). Indeed, serial electron microscopic reconstruction of dendritic spines in mouse cortex reveals that each spine contacts an average of nine different axons, while only forming a synapse with one (*Kasthuri et al., 2015*). Since only a small fraction of the possible connections between neurons exist, how do neurons decide whether to accept or reject potential synaptic contacts? Despite the importance of this question, molecular mechanisms that might enable neurons to specify and control the number of contacts made remain largely unknown.

One way to generate the sparse set of contacts found in the brain would be to regulate the number of contacts each neuron is capable of making relative to its nearby neighbors. The observation that synapse density varies greatly between neighboring neurons and does not correlate to local axon density suggests that neurons may physically interview potential synaptic partners, using molecular cues that control synapse density (*Brown and Hestrin, 2009*; *Chklovskii et al., 2004*; *Druckmann et al., 2014*; *Stepanyants and Chklovskii, 2005*). Consistent with a role for such molecular cues, synapse development proceeds normally in the absence of neuronal activity or even neurotransmitter release (*Augustin et al., 1999*; *Sando et al., 2017*; *Sigler et al., 2017*). By defining how potential synaptic inputs are distributed between neighboring neurons as they develop, molecular cell-contact driven mechanisms could both limit the overall local connectivity and regulate local variations in synapse density.

Numerous trans-synaptic cell adhesion interactions serve as mediators of synapse formation and maturation and likely provide for specificity between synaptic partners (*Biederer and Stagi, 2008*; *Dalva et al., 2007*; *Lai and Ip, 2009*; *McMahon and Díaz, 2011*). Trans-synaptic interactions can coordinate the wiring of individual synapses, but recent evidence suggests that trans-synaptic signaling may also control how many synapses a neuron receives in relation to nearby neurons. Indeed, differences in the expression of trans-synaptic adhesion proteins and BDNF/TrkB can control the local distribution of synaptic inputs (*Bian et al., 2015*; *English et al., 2012*; *Kwon et al., 2012*; *McClelland et al., 2010*). These studies suggest that similar to activity-dependent competition in the developing brain (*Katz and Shatz, 1996*), neurons may use molecular cues to compete with one another for synapses. While each of these studies provides some evidence for a molecular competition, they fail to demonstrate that a molecular competition occurs, since specific winning and losing neurons cannot be identified in complex in vivo or in vitro systems.

One molecule linked to a competitive mechanism for synapse development is eB3. Relative, but not absolute levels of eB3 expression regulate excitatory synapse density in cortical neurons, without affecting inhibitory synapses (*McClelland et al., 2010*). Thus, $Efnb3^{-/-}$ cortical neurons have normal synapse density in vitro and in vivo but generate fewer synaptic contacts when co-cultured with wild type neurons at a 1:10 ratio. Likewise, wild type neurons co-cultured with eB3 null neurons at a 1:10 ratio have higher than normal synapse density (*McClelland et al., 2010*) . Consistent with these findings, shRNA knockdown of eB3 using low-efficiency transfection methods also reduces synapse density in vitro (*McClelland et al., 2010*).

The above results led us to propose a model in which post-synaptic eB3 acts as a competitive signal that enables neurons with higher eB3 levels to generate more synapses. For what are neurons competing? Ephrin-B3 can regulate synapse density through a trans-synaptic interaction with pre-synaptic EphB2 (*McClelland et al., 2010*), suggesting that neurons might compete for EphB2. In this model, higher eB3 expression gives neurons an advantage to more effectively compete for binding of EphB2 found in axons. Based on this neuron-neuron competition model we predict that: 1) relative differences in eB3 expression levels between two neurons should generate a winning (more eB3) and a losing (less eB3) neuron with higher and lower synaptic density respectively, without altering the total number of synapses received by the two neurons; and 2) blocking or preventing interactions between eB3 and EphB2 should nullify the competitive signaling, leading to equal synapse density regardless of the level of eB3 in each post-synaptic neuron.

Here, using a combination of approaches, including a two-neuron microisland cortical neuron culture system that allows assignment of winners and losers in a competition for synapses, we demonstrate that neurons use eB3 to compete for pre-synaptic EphBs. Using the Mosaic Analysis with Double Markers (MADM) genetic mouse model we show that relative levels of ephrin-B3 regulate dendritic spine density in layer 5 and 6 pyramidal neurons in vivo. Remarkably, control of local synapse density by eB3 persists in the absence of action potential-driven activity, suggesting that eB3-mediated competition functions independently of activity-dependent mechanisms in cultured cortical pyramidal neurons. Together these results define eB3 as a trans-synaptic signal that is not required for synapse formation but instead controls the local distribution of a limited pool of synapses by competing for EphBs.

## Results

### Ephrin-B3 does not regulate synapse density in a non-competitive environment

Our previous findings in complex cortical neuron cultures and in vivo suggest a model in which post-synaptic eB3 regulates synapse density through a molecular contest for pre-synaptic EphBs (*McClelland et al., 2010*). However, the complexity of these systems precludes testing of the hypothesis that eB3 directs a molecular competition for synaptic inputs between adjacent neurons. To address whether relative differences in eB3 expression between identified neurons control a competition for synaptic inputs, we utilized a microisland cortical culture system (*Allen, 2006*; *Tarsa and Goda, 2002*) (*Figure 1—figure supplement 1*). We generated two types of islands: one containing a single neuron, and a second containing two neurons. Unlike previous studies in vivo, in brain slice, or in culture (*Bian et al., 2015*; *English et al., 2012*; *Kwon et al., 2012*; *McClelland et al., 2010*), each microisland is a closed system, enabling neurons to be isolated without the opportunity to compete (single-neuron system), or providing for direct competition between identifiable neurons (two-neuron system). Moreover, the simplified nature of the microisland system allows the essential aspect of a competition to be observed: a winner and a loser. We expect that if eB3 directs a contest for synapses between neighboring neurons, eB3 knockdown in single-neuron microislands should have no effect on synapse density since competition cannot occur. In contrast, knockdown of eB3 in one neuron within a two-neuron microisland should reduce this neuron's ability to compete. Thus, if eB3 regulates a competition for synapses, these two-neuron doublets should generate a 'loser', the neuron with reduced eB3, with lower synapse density and a 'winner', the untransfected neuron, with higher synapse density, without changing the total density of synapses in the microisland.

To test these hypotheses, we first generated single-cortical neuron microisland cultures and knocked down eB3 expression with a lentivirus containing a previously characterized eB3 shRNA that selectively reduces eB3 expression by at least 50% (*Figure 1*) (*McClelland et al., 2010*). We then quantified synapse density by fixing and immunostaining cultures for pre- and post-synaptic markers (vGlut1 and PSD-95, respectively), a method which we have shown detects decreases in synapse number following eB3 knockdown as confirmed by a concomitant reduction in mEPSC frequency (*McClelland et al., 2010*). In these single-neuron microislands, synapse density should be unaffected by changes in eB3 expression level. As expected, synapse density in neurons transduced with GFP and either eB3 shRNA or control virus were not significantly different (*Figure 1c,d*). Thus, in conditions where cell-cell competition cannot occur, eB3 does not regulate synapse density or the ability of neurons to form synapses.

### PSD-95-GFP reliably localizes to synapses and does not alter synapse number

Knockdown of eB3 or mixing eB3 null neurons with wild-type neurons at a 1:10 ratio results in reduced synapse density in cells within complex neuronal cultures (*McClelland et al., 2010*). To test whether this results from a failure of neurons expressing low levels of eB3 to compete for synapses, the levels of eB3 in one neuron within a two-neuron microisland were reduced and the effects on synapse density of both neurons in the microisland determined. In order to distinguish synaptic contacts onto the transfected neuron from those onto the transfected neuron within microisland doublets, we developed a Gateway compatible expression construct (Fisher Scientific) expressing PSD-95-GFP and cell filling tdTomato from two distinct mammalian promoters. In addition, this vector expressed either eB3 shRNA (pFUGW-tdTomato-eB3.2) or pSuper control (pFUGW-tdTomato-pSuper) from a separate promoter. Using this construct, we found that PSD-95-GFP reliably localized to post-synaptic sites and could be effectively used to determine the density of synapses onto transfected neurons (*Figure 2a–c*). Importantly, with our transfection conditions the expression of PSD-95-GFP did not alter the density of synapses onto neurons in either single- or two-neuron microislands (*Figure 2d* and *Figure 3*). It should be noted that although expression of PSD-95-GFP to mark post-synaptic specializations did not alter synapse density in our culture system, any potential increase in synapse number caused by PSD-95-GFP expression would only be expected to lessen the effect of eB3 shRNA expression within competitive doublets, since eB3 shRNA and PSD-95-GFP

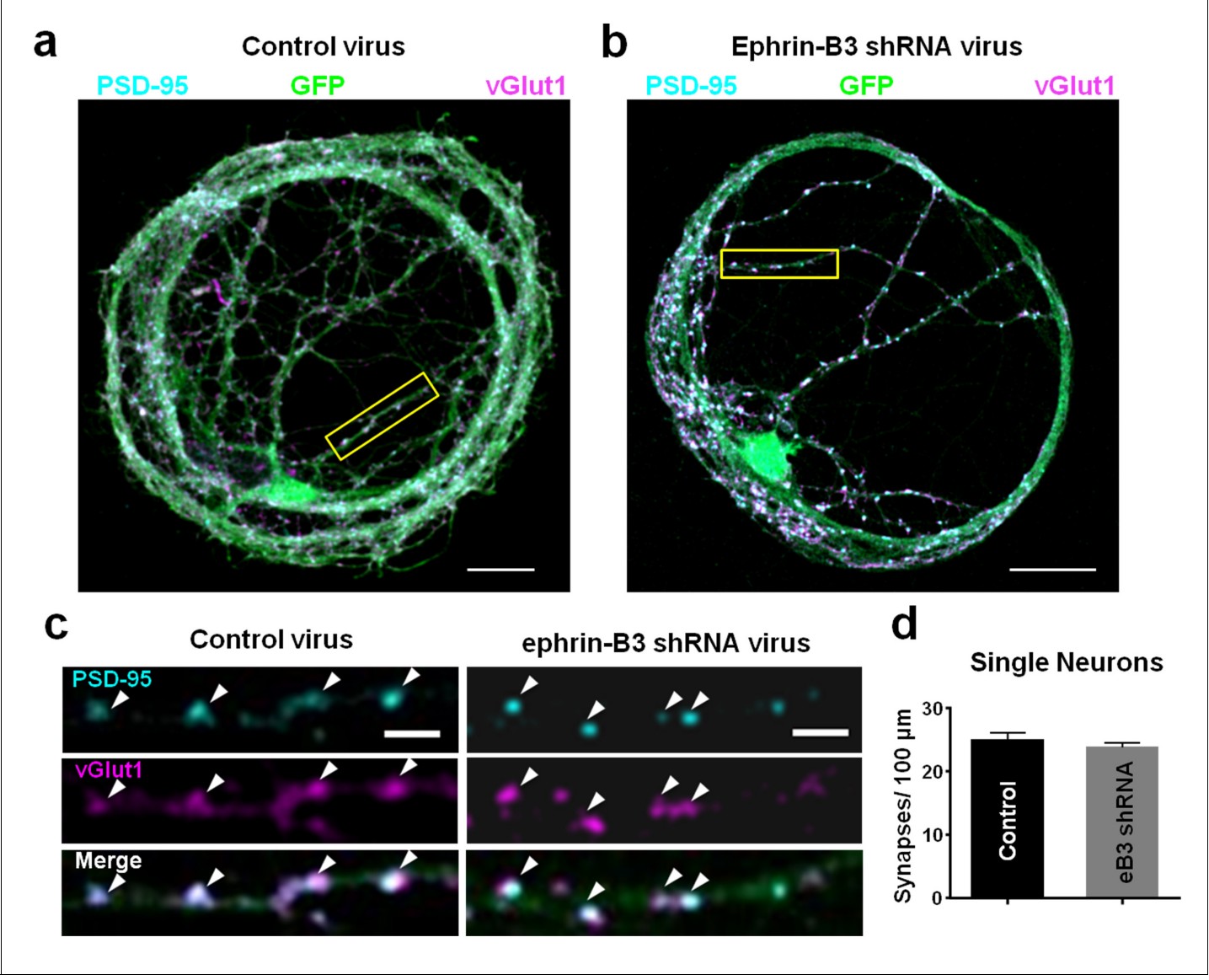

**Figure 1.** Knockdown of ephrin-B3 does not alter synapse density in single-neuron microislands. (a and b) Representative images of dendrites from single-neuron microislands transduced with control (pSuper) and eB3 shRNA viruses and immunostained at DIV21 for vGlut1 and PSD-95. Scale bar, 20 µm. (c) Higher magnification images of boxed regions shown in a and b. Arrowheads indicate colocalized vGlut1 and PSD-95 puncta. Scale bar, 5 µm. (d) Quantification of synapse density in single neuron microislands (Control, n=27; eB3 shRNA, n=47; t(72)=0.9390, p=0.3509, two-tailed Student's t-test) transduced with the indicated viruses.

DOI: https://doi.org/10.7554/eLife.41563.002

The following source data and figure supplement are available for figure 1:

**Source data 1.** Knockdown of ephrin-B3 does not alter synapse density in single-neuron microislands.
DOI: https://doi.org/10.7554/eLife.41563.003
**Figure supplement 1.** Schematic of microisland experimental design.
DOI: https://doi.org/10.7554/eLife.41563.004

were expressed in the same cell. Thus, this approach is well suited to examine whether eB3 regulates the ability of individual neurons to compete for pre-synaptic contacts. Consistent with the results of eB3 knockdown using lentiviral transduction (*Figure 1*), expression of eB3 shRNA combined with PSD-95-GFP did not affect synapse density in single-neuron microislands (*Figure 2d*).

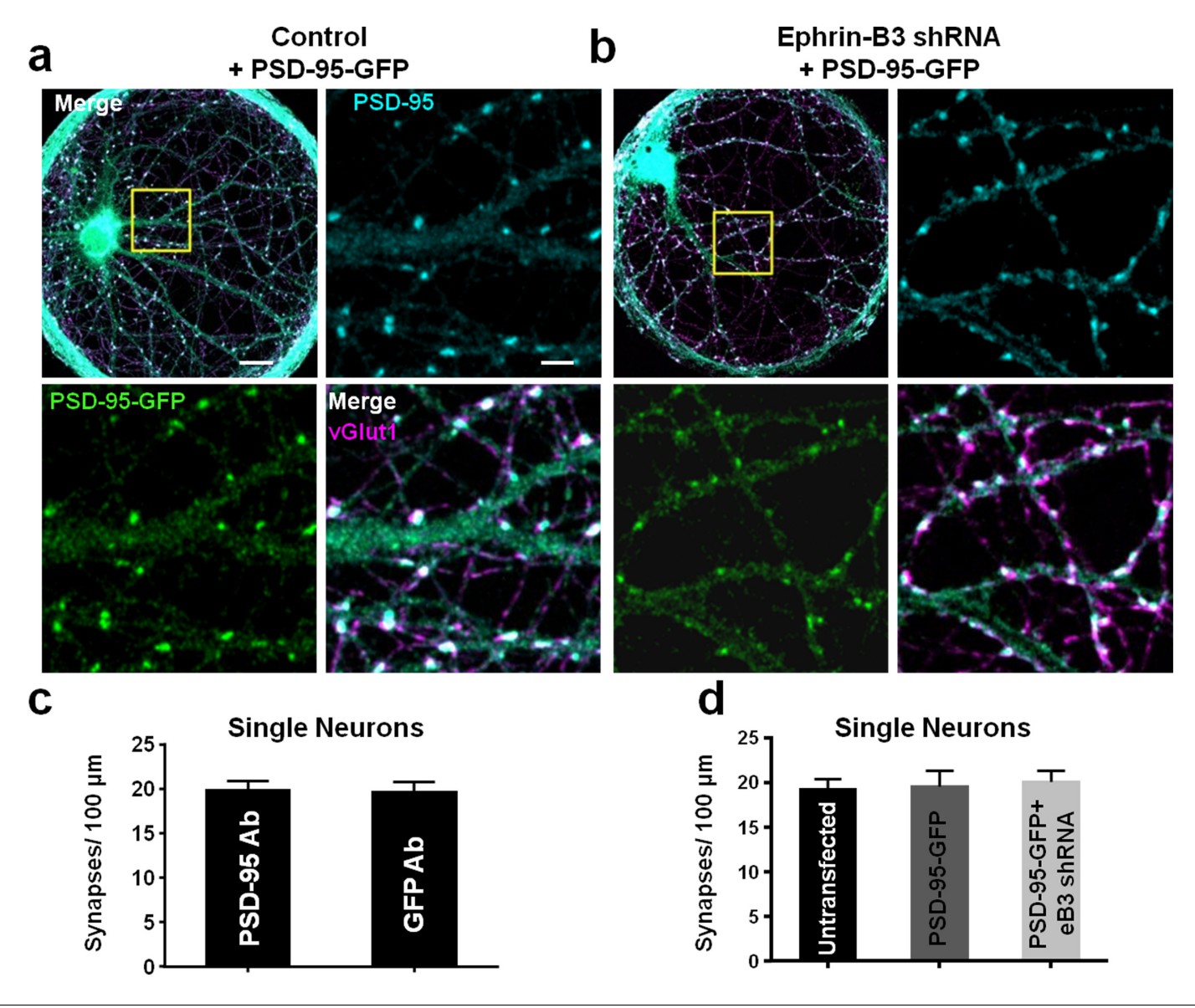

**Figure 2.** PSD-95-GFP localizes to synapses and does not alter synapse density. (**a and b**) Representative images of single-neuron microislands transfected with the indicated constructs and immunostained for GFP (for PSD-95-GFP), PSD-95 and vGlut1. Boxed regions are shown in enlarged insets. Scale bars represent 20 µm in merged images and 5 µm in insets. (**c**) Comparison of synapse density quantification in single neuron microislands using PSD-95 or GFP immunostaining for PSD-95-GFP (n = 18 cells; t(34)=0.08148, p=0.9355, two-tailed Student's t-test). (**d**) Quantification of synapse density in single neurons transfected with the indicated constructs using PSD-95 and vGlut1 immunostaining (Untransfected, n = 20; PSD-95-GFP, n = 18; PSD-95-GFP + eB3 shRNA, n = 18; F(2,53)=.1008, p=0.9043, one-way ANOVA).

DOI: https://doi.org/10.7554/eLife.41563.005

The following source data is available for figure 2:

**Source data 1.** PSD-95-GFP localizes to synapses and does not alter synapse density.

DOI: https://doi.org/10.7554/eLife.41563.006

## Ephrin-B3 directs a molecular competition for synapses in microislands

We next generated control microislands in which we expect that neurons would have no consistent competitive advantage in forming synapses by combining two neurons with endogenous levels of eB3 expression in a single microisland (*Figure 3a,b*). To do this, we transfected neurons by electroporation in suspension with our PSD-95-GFP control construct (pFUGW-tdTomato-pSuper). Two-cell

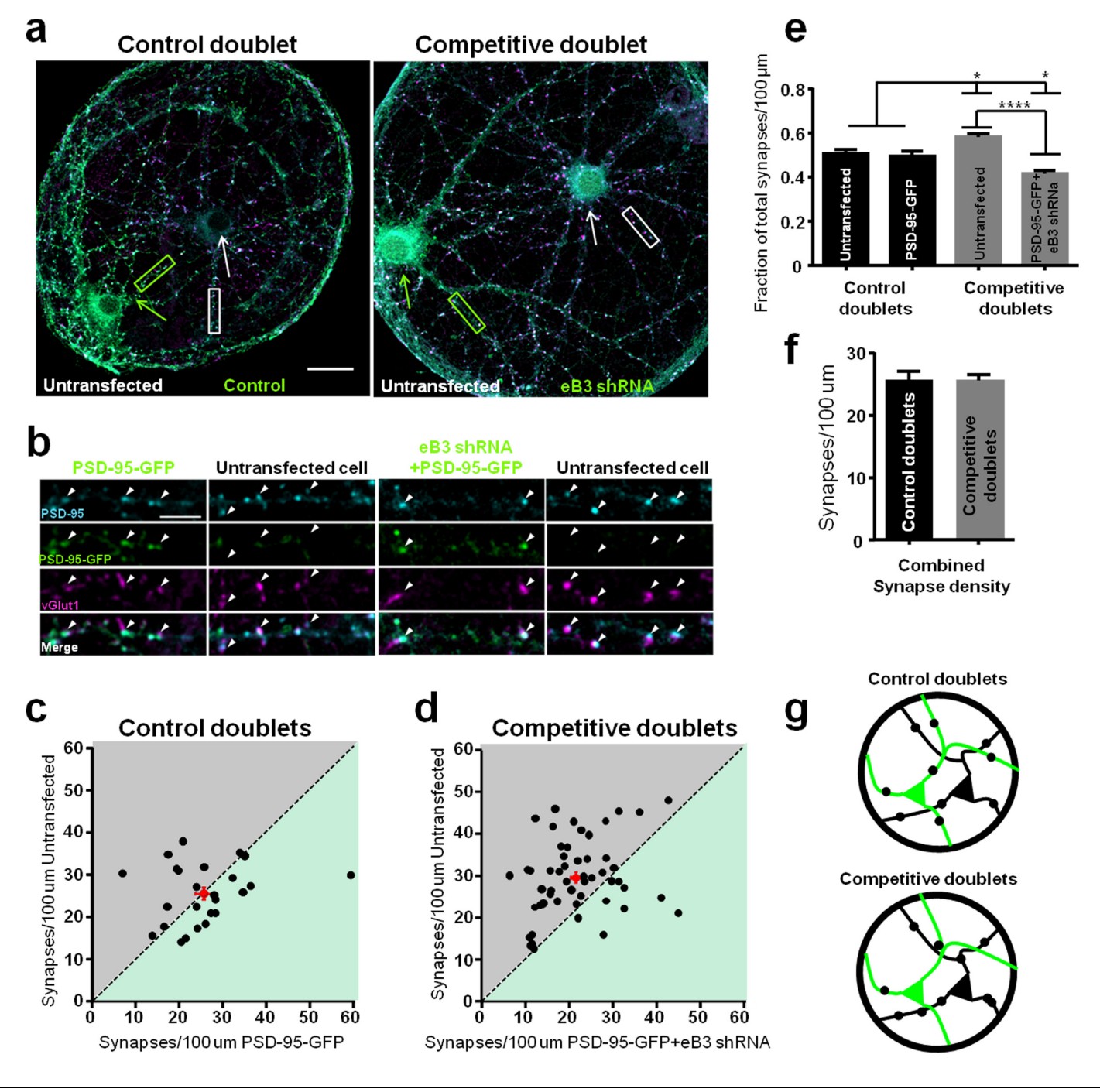

**Figure 3.** Synapse density is regulated by an ephrin-B3-dependent competitive mechanism. (**a**) Representative images of control and competitive doublets. Neurons were immunostained for vGlut1, PSD-95 and GFP (for PSD-95-GFP) at DIV21. Arrows indicate untransfected (white arrow) and transfected (green arrow) neurons. Scale bar, 20 μm. (**b**) Higher magnification images of the boxed regions shown in a. Arrowheads indicate examples of colocalized puncta. Scale bar, 5 μm. (**c and d**) Scatter plots in which the x and y coordinates of each point respectively represent synapse density in transfected and untransfected neurons within a doublet pair. The red point in each graph represents the average of all data points. (**e**) Quantification of the fraction of total synapse density on neurons within control and competitive doublets (control doublets, n=25 for untransfected and PSD-95-GFP cells; competitive doublets, n=53 for untransfected and eB3 shRNA cells; F(3,152)=23.12, p<0.0001, one-way ANOVA *P<0.05,****P<0.0001 Tukey's post hoc. (**f**) Quantification of total synapse density in doublets (control doublets, n=25; competitive doublets n=53; t(76)=.01448, p=0.9882, two-tailed Student's t-test). (**g**) Schematic of the microisland doublet experiment. Black and green neurons indicate untransfected and transfected neurons, respectively. Synapses are represented as black dots.

*Figure 3 continued on next page*

*Figure 3 continued*

DOI: https://doi.org/10.7554/eLife.41563.007

The following source data is available for figure 3:

**Source data 1.** Synapse density is regulated by a competitive mechanism.

DOI: https://doi.org/10.7554/eLife.41563.008

microislands were generated when one transfected and one untransfected neuron landed on the same island as the neurons were plated. At day in vitro (DIV) 21 we quantified synapse density in two-neuron microislands by immunostaining for synaptic markers (*Figure 3b*). We have shown previously that expression of eB3 varies between neurons and eB3 levels positively correlate with synapse density (*McClelland et al., 2010*). Therefore, we expected that neither the transfected or untransfected cell would consistently have more synapses, but that one neuron in each pair of cells might have more synapses than the other. To capture variation between individual cells, we plotted synapse density in untransfected vs. control (PSD-95-GFP) transfected neurons on a scatter plot such that each data point represents two cells within a single microisland pair (*Figure 3c*). Synapse density was typically higher in one neuron within each pair (*Figure 3c*, dashed line represents equal synapse density), but there was no bias in whether the transfected or untransfected neuron won. Accordingly, untransfected and transfected cells in these control doublets received an equal proportion of synapses within the doublets on average, indicating that synapses were equally distributed among neurons.

Next, we created a situation with reliable differences in eB3 levels with the expectation that a consistent winning and losing neuron would emerge. Neurons were transfected with our eB3 knockdown construct (pFUGW-tdTomato-eB3.2) and microislands were created by transfected and untransfected neurons that landed on individual islands (*Figure 3a,b*). In competitive doublets where eB3 was knocked down in one of the two neurons, eB3 shRNA transfected cells received a significantly lower proportion of total synaptic contacts than untransfected cells, and a significantly lower proportion of synaptic contacts than transfected and untransfected cells in control doublets (*Figure 3d,e*). Likewise, untransfected cells received a higher proportion of synapses within competitive doublets, and a higher proportion of synapses than untransfected and transfected cells in control doublets (*Figure 3d,e*). Thus, in a single two-neuron microisland, a higher relative level of eB3 in a cell provides a competitive advantage, generating a winner in a competition for synaptic inputs. Neurons within microisland doublets have a broad range of synapse densities (*Figure 3c,d*). Despite this variation, examination of competitive doublets shows that neurons with endogenous levels of eB3 (untransfected) consistently emerged as winners (83% of doublets). In contrast, control transfected cells were winners much less often (83% vs. 44%, Competitive vs. Control p=0.001, Fisher's exact test). There was a ~ 35% increase in synapse density in untransfected neurons compared to eB3 knockdown neurons within the same competitive doublet (mean difference, untransfected cell-transfected cell, n = 53 doublets, 8.05 ± 1.48 synapses/100 μm, p<0.0001, one-sample t-test), while no such difference was observed in control doublets (n = 25 doublets, 0.15 ± 2.18 synapses/100 μm, p=0.994, one sample t-test). Total synapse density in competitive doublets in which one neuron expressed eB3 shRNA was unchanged from that of control doublets (*Figure 3f*). This indicates that knockdown of eB3 did not alter the overall number of synapses that the pair of neurons could form. Rather, knockdown of eB3 results in a redistribution of synapses onto neurons with higher eB3 expression (winners) at the expense of neurons that were transfected with eB3 shRNA (losers). These results suggest that eB3 functions as a cell-cell competition signal that controls how a limited pool of synaptic inputs is distributed between neurons (*Figure 3g*).

## Ephrin-B3 competes for EphBs

What might be controlling the pool of synaptic inputs neurons can receive? We have previously demonstrated that knockdown of pre-synaptic EphB2 completely blocks the ability of eB3 to induce pre-synaptic differentiation in a heterologous cell synapse induction assay, suggesting that pre-synaptic EphB2 is required in this system (*McClelland et al., 2010*; *Scheiffele et al., 2000*). Consistent with this finding, biochemical fractionation revealed that EphB2 is localized to pre-synaptic active zones, though not exclusively (*Bouvier et al., 2008*). We, therefore, hypothesized that neurons are

competing for pre-synaptic EphBs. In this model, higher levels of eB3 expression would enable a cell to out-compete neighboring cells for axonal contacts expressing EphBs, leading to higher synapse density. This model predicts that preventing the interaction between eB3 and EphBs should nullify the competitive advantage conferred by higher levels of eB3 expression, preventing these cells from competing for EphB-expressing axons and thereby rescuing synapse density in neurons with reduced eB3 to control levels. To block the interaction between eB3 and endogenous EphBs, we applied unclustered EphB2 ectodomain that binds ephrin-Bs but does not activate ephrin-B signaling (*San Miguel et al., 2011*). In the context of eB3 knockdown, we expected that EphB2 ectodomain would functionally eliminate differences in eB3 expression between cells expressing eB3 shRNA and the surrounding untransfected cells by preferentially binding to cells expressing wild-type levels of eB3. To confirm that binding of EphB2 ectodomain is reduced in cells expressing eB3 shRNA, we first transfected E17-18 rat cortical neurons at DIV0 with EGFP and either eB3 shRNA or control vector (*McClelland et al., 2010*). We then applied EphB2 ectodomain for 45 min at DIV10, fixed the cultures, and stained for exogenously applied EphB2 ectodomain (*Figure 4—figure supplement 1a,b*). As expected, EphB2 ectodomain puncta density and size were reduced in neurons expressing eB3 shRNA compared to control neurons (*Figure 4—figure supplement 1c,d*). In addition, we confirmed that unclustered EphB2 ectodomain does not activate ephrin-B signaling (*Figure 4—figure supplement 1e*).

We next tested our hypothesis that applying exogenous EphB2 ectodomain in this manner would nullify the competitive advantage of high eB3 expression, rescuing synapse density in neurons expressing eB3 shRNA. After transfecting neurons with GFP and either eB3 shRNA or control vector as described above, we treated cultures from DIV3-10 with EphB2 ectodomain, fixed the neurons at DIV10 and determined synapse density by immunostaining for PSD-95 and vGlut1. Knockdown of eB3 in these complex cortical neuron cultures significantly reduced synapse density (*Figure 4a,b*). Remarkably, treatment with unclustered EphB2 ectodomain rescued this reduction in synapse density to control levels (*Figure 4a,b*). Consistent with the competition model, the rescue of synapse density by treatment with EphB2 ectodomain was dose-dependent (*Figure 4b*). The effects of EphB2 ectodomain were specific to cultures containing neurons with reduced eB3 expression, since treatment of control-transfected neurons with EphB2 ectodomain had no effect on synapse density (*Figure 4b*). These findings suggest that control of synapse density by eB3 relies on relative differences in eB3 expression, which allow cells to compete for binding of EphBs. As eB3 requires EphB2 in axons to initiate pre-synaptic differentiation, eB3 likely allows cells to compete for axonal EphB2 (*McClelland et al., 2010*).

We asked whether we could prevent neurons in competitive microislands from winning or losing by adding unclustered EphB2 ectodomain. Control and competitive two-neuron microislands were generated as described above and treated with EphB2 ectodomain from DIV3-21 to block the competition for EphBs (*Figure 4—figure supplement 2*). Unclustered EphB2 ectodomain treatment did not affect synapse density in control doublets (EphB2 ectodomain treated doublets, n = 14 neurons, 22.4 ± 1.5 synapses/100 μm vs. untreated doublets, n = 50 neurons, 25.6 ± 1.2 synapses/100 μm, t (62)=1.305, p=0.1966, two-tailed Students t-test). However, EphB2 ectodomain treatment completely blocked competition in competitive microislands, preventing both the increase in the winning (untransfected) and losing (eB3 shRNA transfected) neurons (*Figure 4—figure supplement 2c,d*). These findings indicate that eB3 likely establishes synapse density through local neuron-neuron competition for pre-synaptic EphBs.

## Relative levels of ephrin-B3 determine spine density in vivo

To further characterize the role of eB3 in the competition for synapses in vivo, we sought a genetic approach. To this end, we generated eB3 Mosaic Analysis with Double Markers (MADM) mice (*Figure 5—figure supplement 1*) (*Espinosa et al., 2009*; *Hippenmeyer et al., 2010*; *Zong et al., 2005*). This model allowed us to study competitive interactions within a genetically defined population of WT (Wild type) and *Efnb3$^{-/-}$* neurons derived from single animals, identifiable by expression of different fluorescent markers. By crossing MADM-11 mice, which contain the MADM TG and GT cassettes on chromosome 11 (*Hippenmeyer et al., 2010*) to a previously characterized transgenic mouse containing a null version of the *Efnb3* gene on chromosome 11 (*Hruska et al., 2015*; *McClelland et al., 2010*; *Yokoyama et al., 2001*), we generated mosaic eB3 knockout MADM mice. We generated two different eB3 MADM lines: one with Emx1-Cre which drives expression of Cre

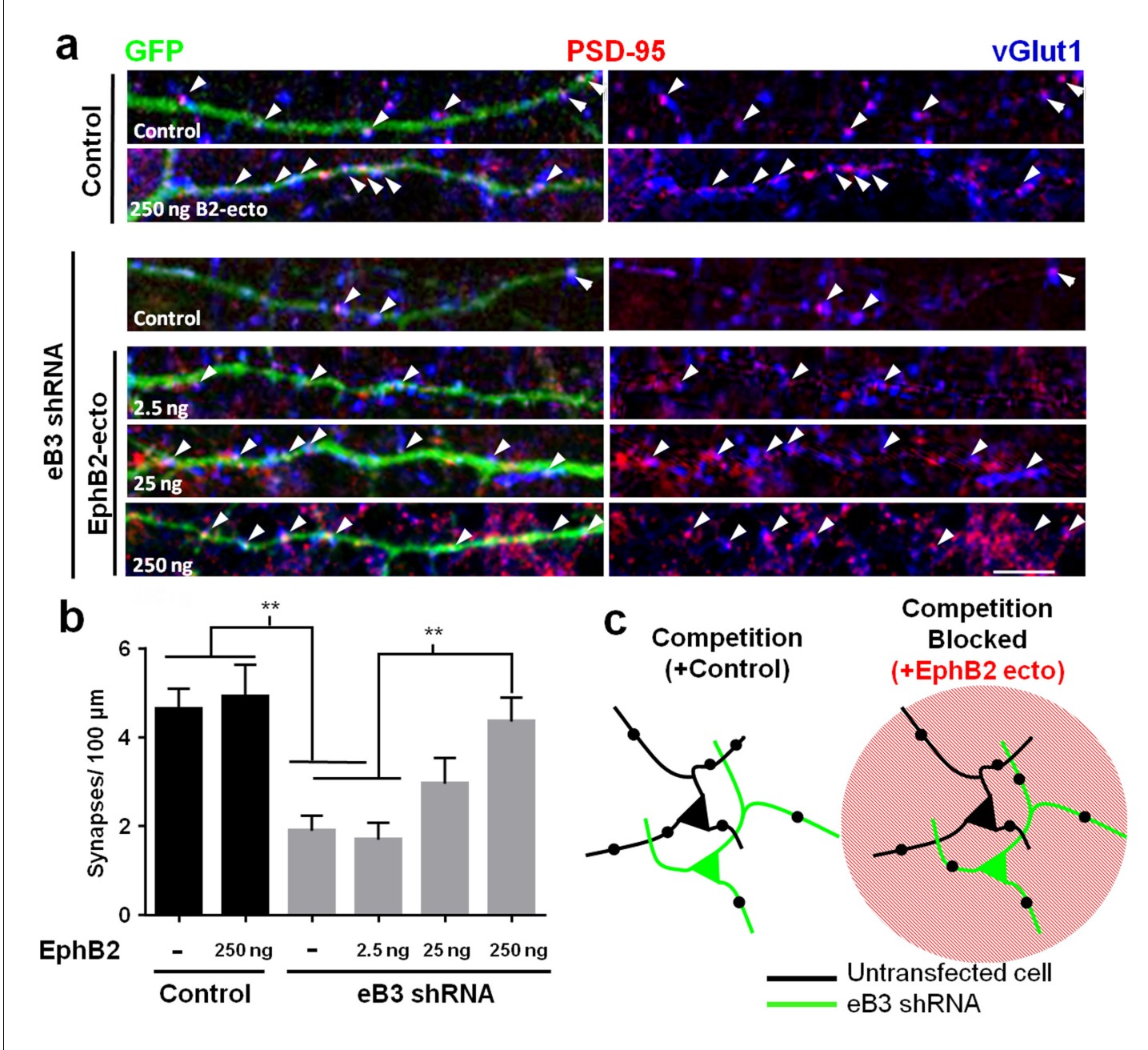

**Figure 4.** Effects of ephrin-B3 knockdown are rescued by unclustered EphB2 ectodomain treatment. (a) DIV10 cortical neurons transfected with the indicated constructs along with GFP and treated with unclustered EphB2 ectodomain (EphB2-ecto). Arrowheads indicate colocalized PSD-95 (red) and VGlut1 (blue) puncta along GFP-labeled dendrite. (b) Quantification of colocalized PSD-95 and VGlut1 puncta density (Control/ctrl, n = 22; ctrl/B2-Fc 250 ng n = 19; eB3 shRNA/ctrl, n = 38; eB3 shRNA/B2-Fc 2.5 ng, n = 25; eB3 shRNA/B2-Fc 25, ng n = 22; eB3 shRNA/B2-Fc 250 ng, n = 26; F (5,146) =8.454, p<0.0001, one-way ANOVA, **p<0.01 (c) Model of the effect of unclustered EphB2 ectodomain treatment on adjacent neurons within complex cultures. Black dots represent synapses.

DOI: https://doi.org/10.7554/eLife.41563.009

The following source data and figure supplements are available for figure 4:

**Source data 1.** Effects of ephrin-B3 knockdown are rescued byEphB2 ectodomain.

DOI: https://doi.org/10.7554/eLife.41563.010

**Figure supplement 1.** Validation of unclustered EphB2 ectodomain treatment method.

DOI: https://doi.org/10.7554/eLife.41563.011

**Figure supplement 1—source data 1.** Validation of EphB2 ectodomain treatment method.

*Figure 4 continued on next page*

*Figure 4 continued*

DOI: https://doi.org/10.7554/eLife.41563.012

**Figure supplement 2.** Unclustered EphB2 ectodomain blocks competition in microislands.

DOI: https://doi.org/10.7554/eLife.41563.013

**Figure supplement 2—source data 1.** EphB2 ectodomain blocks competition in microislands.

DOI: https://doi.org/10.7554/eLife.41563.014

recombinase in progenitors of excitatory forebrain neurons (*Beattie et al., 2017*; *Gorski et al., 2002*), and another line with Nestin-Cre that drives expression of Cre recombinase in all neural progenitors but results in much sparser labeling (Nestin-spCre) (Materials and methods) (*Hippenmeyer et al., 2010*; *Zong et al., 2005*). In both lines, sparse recombination generated fluorescently labeled neurons in cortex with known eB3 genotype. In the Wild type MADM (control MADM) all labeled and unlabeled cells are WT. While in the eB3 mosaic MADM mice, tdTomato +cells are WT, EGFP +cells are *Efnb3$^{-/-}$*, while double labeled tdTomato+/EGFP+ (yellow) cells and unlabeled cells are *Efnb3$^{+/-}$* (*Figure 5—figure supplement 1*).

We next sought to determine whether eB3 controls spine number in vivo by comparing the density of dendritic spines on cortical pyramidal neurons of WT (tdTomato+) and *Efnb3$^{-/-}$* (EGFP+) neurons within the mosaic eB3 MADM mouse. For this experiment, we chose to use our Nestin-spCre MADM line, which exhibits sparse labeling, facilitating spine density quantification. Since labeling in MADM mice is sparse, each WT neuron should have a competitive advantage over the surrounding unlabeled *Efnb3$^{+/-}$* neurons, resulting in an elevated spine density in WT tdTomato + neurons. In contrast, each *Efnb3$^{-/-}$* neuron should be at a competitive disadvantage and receive fewer synapses, resulting in a lower spine density in *Efnb3$^{-/-}$*EGFP + neurons. Finally, double labeled *Efnb3$^{+/-}$* (EGFP +/tdTomato+) neurons should exhibit spine densities in between that of WT and *Efnb3$^{-/-}$* cells.

We previously generated and characterized an antibody that recognizes a specific phosphorylated serine residue (serine 332) near the C terminus of eB3. In brain sections, we observed high immunoreactivity in large apical dendrites of sub-granular layer 5 and 6 pyramidal neurons (*Hruska et al., 2015*). We hypothesized that these eB3-expressing cells with thicker apical dendrites may be thick-tufted projection neurons that express the marker CTIP2 (*Chen et al., 2008*; *Leone et al., 2015*). These neurons are known to be larger than callosally projecting (SATB2+) neurons. To test this, we used three color RNAscope in situ hybridization (ISH) to assess the expression of eB3 mRNA (*Efnb3*) in subgranular neurons expressing CTIP2 mRNA (*Bcl11b*) and SATB2 mRNA (*Satb2*) alone or in conjunction (*Anderson et al., 2016*; *Ataman et al., 2016*; *Harb et al., 2016*) (*Figure 5*). Both CTIP2 + and CTIP2+/SATB2 + cells expressed higher levels of *Efnb3* than SATB2 + cells (*Figure 5a,b*). The low level of RNAscope signal for *Efnb3* in SATB2 + neurons was the same as found in *Efnb3$^{-/-}$* mice (*Hruska et al., 2015*; *Yokoyama et al., 2001*) (*Figure 5—figure supplement 2*), suggesting that SATB2 + cells may not express eB3.

Consistent with this, in *Efnb3$^{-/-}$* cells, we found a significant reduction in eB3 signal in CTIP2 + cells that were indistinguishable from *Efnb3* levels in WT SATB2 + neurons, while no additional decrease in eB3 mRNA levels was observed in SATB2 + cells (*Figure 5—figure supplement 2*). Thus, we hypothesized that loss of eB3 would reduce synapse density in CTIP2 + layer 5 and 6 neurons, leaving neighboring SATB2 + neurons unaffected.

To determine whether we could distinguish CTIP2 and SATB2 expressing neurons based on their morphology, we stained control and eB3 MADM brain sections for CTIP2 and SATB2 (*Figure 5c*) (*Alcamo et al., 2008*). Consistent with previous findings, we found that the apical dendrites of CTIP2 + neurons were significantly thicker than those of CTIP2-/SATB2 + neurons, with most of them exceeding 1.6 μm in diameter (CTIP2+, n = 31; CTIP2-/SATB2+, n = 12; 2.04 ± 0.10 vs. 1.57 ± 0.08 μm; t(41)=2.697, p=0.0101, two-tailed Student's t-test) (*Chen et al., 2008*; *Oswald et al., 2013*) (*Figure 5d*). In contrast, most CTIP2-/Satb2 + neurons were less than 1.6 μm in diameter (*Figure 5d*). We next asked whether eB3 expression varied within the population of CTIP2 + thick apical dendrite neurons. Using RNAscope, we found that *Efnb3* expression varied > 4 fold in CTIP2 + cells (*Figure 5—figure supplement 2*). Together with results from RNAscope, these data suggest that within layers 5 and 6, differences in eB3 expression levels might have selective effects in CTIP2 + subcortically projecting neurons with thick apical dendrites.

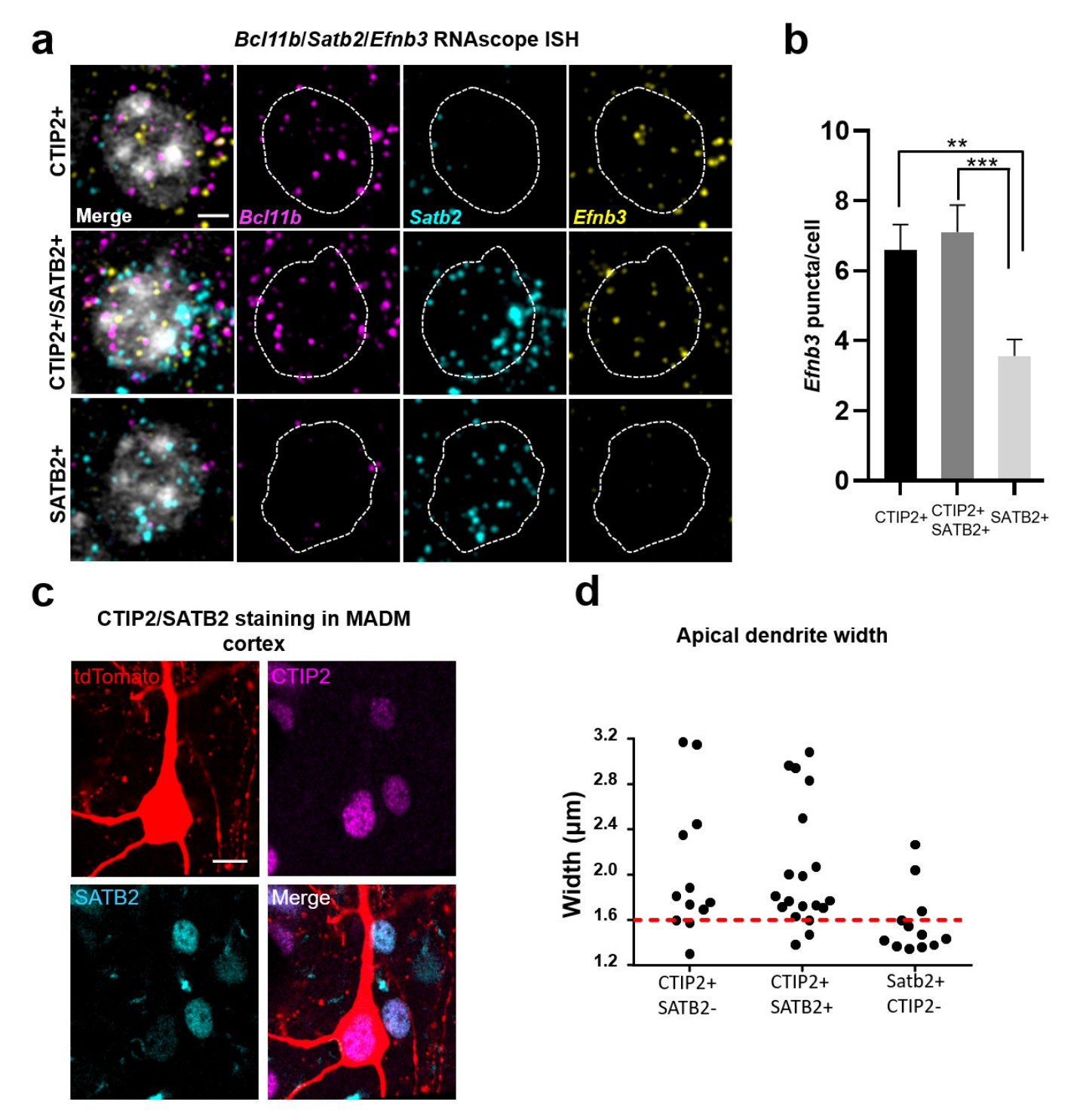

**Figure 5.** *Efnb3* is expressed in CTIP2 +projection neurons. (**a**) Representative cells within WT mouse cortex labeled by RNAscope ISH for CTIP2 mRNA (*Bcl11b*), SATB2 mRNA (*Satb2*), and eB3 mRNA (*Efnb3*). Scale bar, 3 μm. (**b**) Quantification of the number of RNAscope ISH eB3 puncta per cell across the indicated cell types (CTIP2+, n = 72; CTIP2+/SATB2+, n = 69; SATB2+, n = 93; F (2, 231)=9.332, p=0.0001, one-way ANOVA, ***p=0.0004, **p=0.0026 *, Tukey's post hoc.) (**c**) Representative image of a tdTomato+/CTIP2 +cell in MADM cortex co-labeled with CTIP2 and SATB2 antibodies. Scale bar, 10 μm. (**d**) Analysis of apical dendrite width in subgranular neurons expressing CTIP2 and/or SATB2. Most neurons expressing CTIP2 have large (>1.6 μm) dendrites.

DOI: https://doi.org/10.7554/eLife.41563.015

*Figure 5 continued on next page*

*Figure 5 continued*

The following source data and figure supplements are available for figure 5:

**Source data 1.** *Efnb3* is expressed in CTIP2 +projection neurons.
DOI: https://doi.org/10.7554/eLife.41563.016
**Figure supplement 1.** Schematic of recombination in eB3 MADM mice.
DOI: https://doi.org/10.7554/eLife.41563.017
**Figure supplement 2.** Controls for RNAscope ISH.
DOI: https://doi.org/10.7554/eLife.41563.019
**Figure supplement 2—source data 1.** Controls for RNAscope ISH.
DOI: https://doi.org/10.7554/eLife.41563.020

To begin to test this, we quantified the density of dendritic spines on the apical dendrites of sub-granular layer 5 and 6 neurons in MADM animals. In control MADM mice, we observed no differences in average spine density between EGFP+, tdTomato+, or EGFP+/tdTomato+ (yellow) cells (*Figure 6—figure supplement 1*). Thus, we grouped these three populations of cells for further analyses. In eB3 MADM mice, no differences in average spine density were observed in cells with thin apical dendrites (<1.6 μm) (*Figure 6c,d*). In contrast, neurons with thick apical dendrites in eB3 MADM mice, WT (tdTomato+) neurons had significantly higher spine density than WT neurons from control MADM mice, *Efnb3*$^{-/-}$ (EGFP+) neurons (~38% increase) and *Efnb3*$^{+/-}$ neurons (~21% increase, *Figure 6a,b*). Consistent with the finding that expression levels of eB3 linearly correlate with synapse density (*McClelland et al., 2010*), the average density of spines on *Efnb3*$^{+/-}$ GFP+/ tdTomato+) neurons was in between that of WT and *Efnb3*$^{-/-}$ neurons (*Figure 6a,b*). Together, with our RNAscope data these results support a model where the relative levels of eB3 in CTIP2 + subgranular neurons functions to competitively regulate spine density.

## Ephrin-B3 controls synapse number in pyramidal neurons of defined genotypes

Ephrin-B3 regulates spine density in a competitive fashion in layer 5 and 6 neurons in eB3 MADM mice. To test whether synapse density is regulated in a similar manner, we used a simplified culture system. We generated cultures of genetically defined neurons by dissociating cortices of eB3 MADM (heterogenotypic cultures) or control MADM (control cultures) mice at P0 and used FACS to isolate EGFP + and tdTomato + cells (*Figure 7a,b*). We used our Emx1-Cre eB3 MADM line for this experiment due to the higher density of labeled cells, which increased the yield of FACS-purified cells.

To maximize the opportunity for competition between WT (tdTomato+) and *Efnb3*$^{-/-}$ (EGFP+) cells, we plated tdTomato + and EGFP + cells at a 1:1 ratio (*Figure 7b,c*). At DIV11, the cultures were fixed and stained for the excitatory synaptic marker vGlut1. We then took an unbiased imaging approach in which a confocal microscope was set to image each culture in its entirety in sequential 150 × 150 μm image fields using image tiling (*Figure 7d*) (Materials and methods). To test whether WT cells exhibit a competitive advantage over nearby *Efnb3*$^{-/-}$ cells for synaptic contacts in MADM cultures, we quantified the density of vGlut1 + synapses onto WT and *Efnb3*$^{-/-}$ cells in each 150 × 150 μm image field, and asked whether WT cells received a disproportionate pool of these synaptic contacts. In Control MADM cultures, in which EGFP + and tdTomato + cells were both WT, we found no significant difference in synapse density between tdTomato + and EGFP + cells (n = 218 fields, 25.2 ± vs 22.6 ± 1.3 vGlut1 puncta/1000 μm$^2$, t(434)=1.463, p=0.1443, two-way Student's t-test) (*Figure 8a,c*). However, in heterogenotypic MADM cultures containing WT (tdTomato+) and *Efnb3*$^{-/-}$ (EGFP+) cells, WT cells exhibited higher synapse density than *Efnb3*$^{-/-}$ cells (n = 108 fields, 32.5 ± 1.6 vs 25.0 ± 1.4, t (214)=3.527, p=0.0005, two-tailed Student's t-test) (*Figure 8b,d*). Moreover, the proportion of fields in which tdTomato + cells received more synaptic contacts was significantly higher than in control cultures (70% vs. 57%, p=0.0216, Fisher's exact test) (*Figure 8c,d*). The total density of synapses was not significantly different between control and heterogenotypic cultures (control cultures, n = 3; heterogenotypic cultures, n = 3; 23.19 ± 1.2 vs. 26.35 ± 4.43 vGlut1 puncta/1000 μm$^2$; t(4)=0.6866, p=0.5301, two-tailed Student's t-test). These results show that eB3

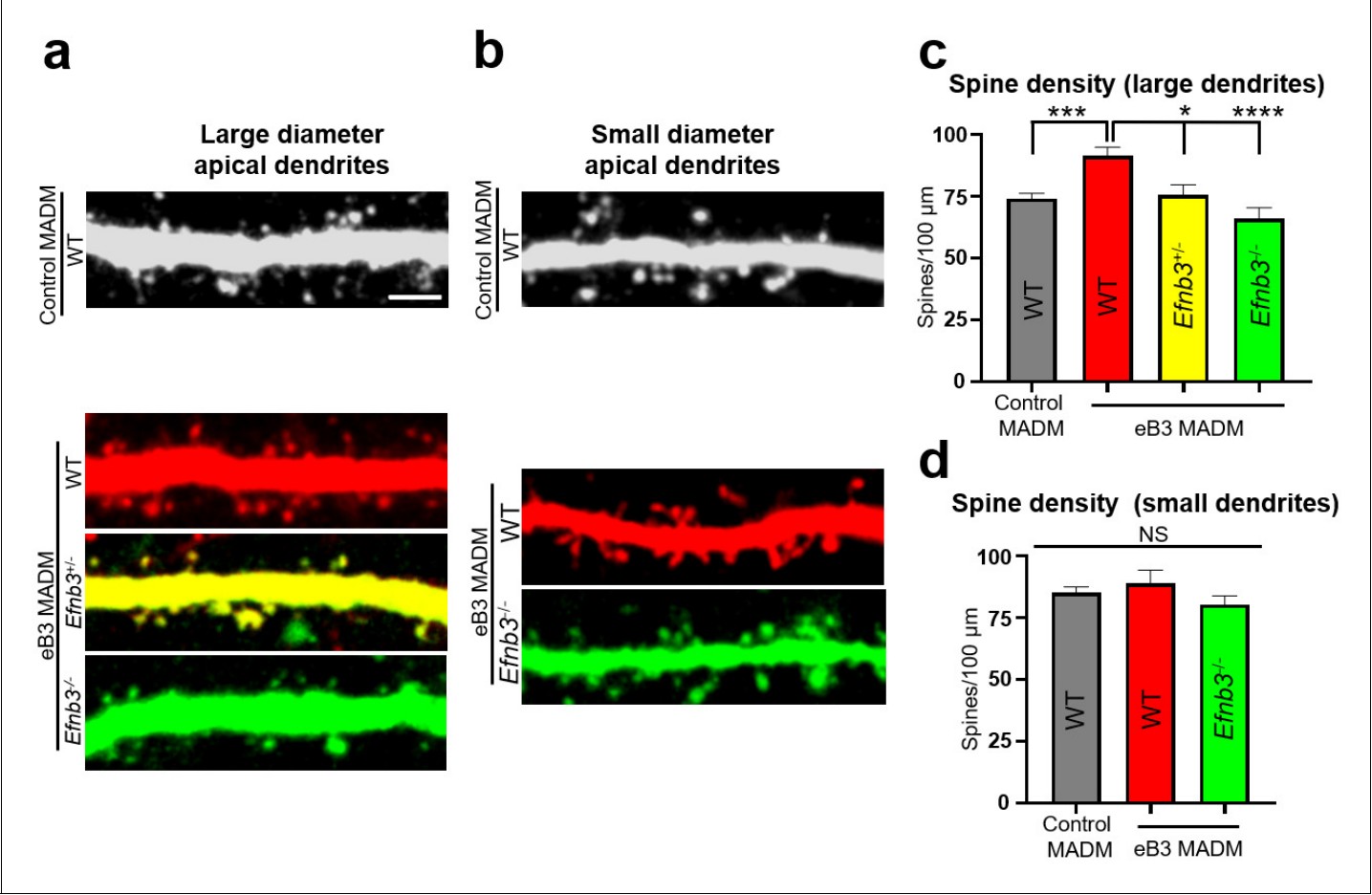

**Figure 6.** Relative levels of ephrin-B3 control spine density in vivo. (a) Representative large diameter (>1.6 μm) dendrites of subgranular pyramidal neurons in control and eB3 Nestin-spCre MADM mice. Scale bar, 3 μm. (b) Representative small diameter (<1.6 μm) dendrites of subgranular pyramidal neurons in control and eB3 Nestin-spCreMADM mice. (c) Quantification of spine density on large diameter dendrites from neurons in control and eB3 MADM mice (control MADM (WT), n=93; eB3 MADM tdTomato+ (WT), n=50; eB3 MADM tdTomato+/EGFP+ (Efnb3+/-), n=24; eB3 MADM EGFP+ (Efnb3-/-), n= 38; $F_{(3, 201)}$= 10.40, p<0.0001, one-way ANOVA, ****p<0.0001, ***p=0.0011 *p<0.05, Tukey's post hoc.) ( d) Quantification of spine density on small diameter dendrites from neurons in control and eB3 MADM mice (control MADM, n=48; eB3 MADM TdTomato+ (WT), n=19; eB3 MADM EGFP+ (Efnb3-/-), n= 33; $F_{(2, 97)}$= 1.283, p=0.2818, one-way ANOVA).

DOI: https://doi.org/10.7554/eLife.41563.021

The following source data and figure supplements are available for figure 6:

**Source data 1.** Relative levels of ephrin-B3 control spine density in vivo.
DOI: https://doi.org/10.7554/eLife.41563.022
**Figure supplement 1.** Equal spine density in control MADM neurons regardless of fluorophore expressed.
DOI: https://doi.org/10.7554/eLife.41563.023
**Figure supplement 1—source data 1.** Spine density WT MADM neurons.
DOI: https://doi.org/10.7554/eLife.41563.024

controls synapse density in a competitive manner in a genetically defined population of WT and *Efnb3*-/- excitatory pyramidal neurons.

## Ephrin-B3-mediated competition does not require neuronal activity

To determine whether competitive interactions between WT and *Efnb3*-/- neurons derived from eB3 MADM mice are dependent on action potential-driven activity. We treated control or heterogenotypic MADM cultures with tetrodotoxin (TTX) to block action potential-driven neuronal activity from DIV3-11, fixed and immunostained the cultures for vGlut1, and determined synapse density as

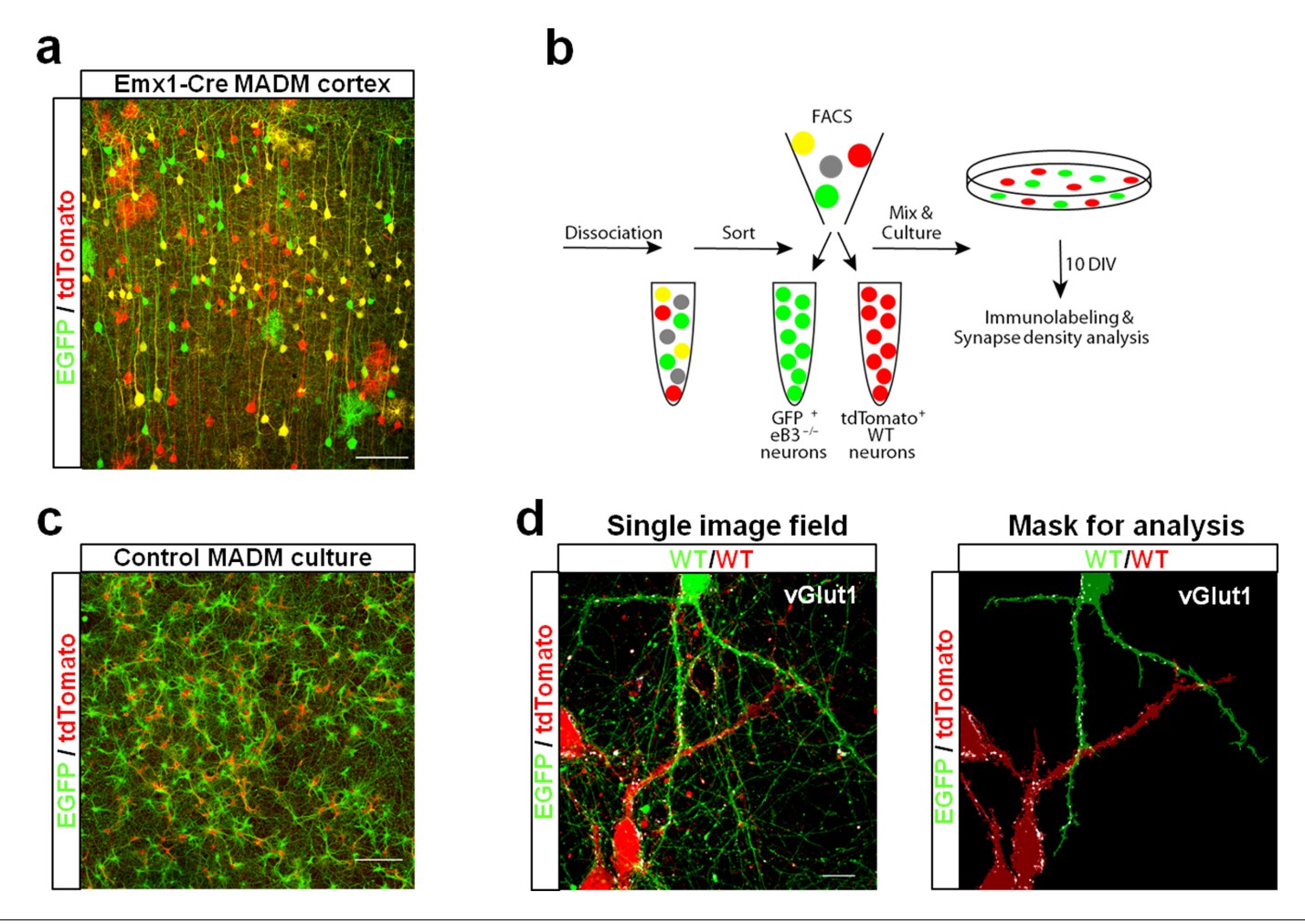

**Figure 7.** Use of ephrin-B3 MADM mice to generate heterogenotypic cultures. (a) Representative image of eB3 MADM (Emx1-Cre line) cortex containing sparse EGFP +and/or tdTomato +expressing cells. Scale bar, 100 μm. (b) Workflow diagram for generating eB3 MADM cultures. (c) Representative image of a control eB3 MADM culture. Scale bar, 300 μm. (d) Representative single image field and processed image mask used for synapse density analysis. Scale bar, 15 μm.

DOI: https://doi.org/10.7554/eLife.41563.025

described above. As expected, synapse density in each image field did not differ between tdTomato + and EGFP + cells in control cultures treated with TTX (n = 166 fields, 25.3.3 ± 1.1 vs 23.1 ± 1.0 vGlut1 puncta/1000 μm$^2$, t(330)=1.433, p=0.1528, two-tailed Student's t-test) (*Figure 9a, c*). In agreement with other work (*Harms and Craig, 2005*; *Sigler et al., 2017*), we did not observe any changes in total synapse density between untreated and TTX-treated control MADM cultures (untreated, n = 3 cultures; TTX treated, n = 3 cultures; 23.19 ± 1.2 vs. 22.57 ± 3.41 vGlut1 puncta/ 1000 μm$^2$; t(4)=0.1715, p=0.8722, two-tailed Student's t-test). If eB3-dependent competition requires action potential-driven neuronal activity, we would expect that in heterogenotypic cultures TTX treatment should block WT (tdTomato+) neurons from winning the competition for synaptic contacts, rescuing the system back to the control condition. In TTX-treated heterogenotypic cultures WT (tdTomato+) cells still received a higher density of vGlut1 + contacts than *Efnb3*$^{-/-}$ (EGFP+) cells (n = 135 fields, 29.8 ± 1.8 vs. 22.9 ± 1.7 vGlut1 puncta/1000 μm$^2$, t(268)=2.772, p=0.006, two-tailed Student's t-test) (*Figure 9b,d*), while the total density of synapses was not different from TTX-treated control MADM cultures (control cultures, n = 3; heterogenotypic cultures, n = 3; 22.57 ± 3.41 vs. 19.97 ± 7.28 vGlut1 puncta/1000 μm$^2$, t(4)=0.3234, p=0.7626, two-tailed Student's t-test). Moreover, in heterogenotypic cultures treated with TTX the proportion of fields with more

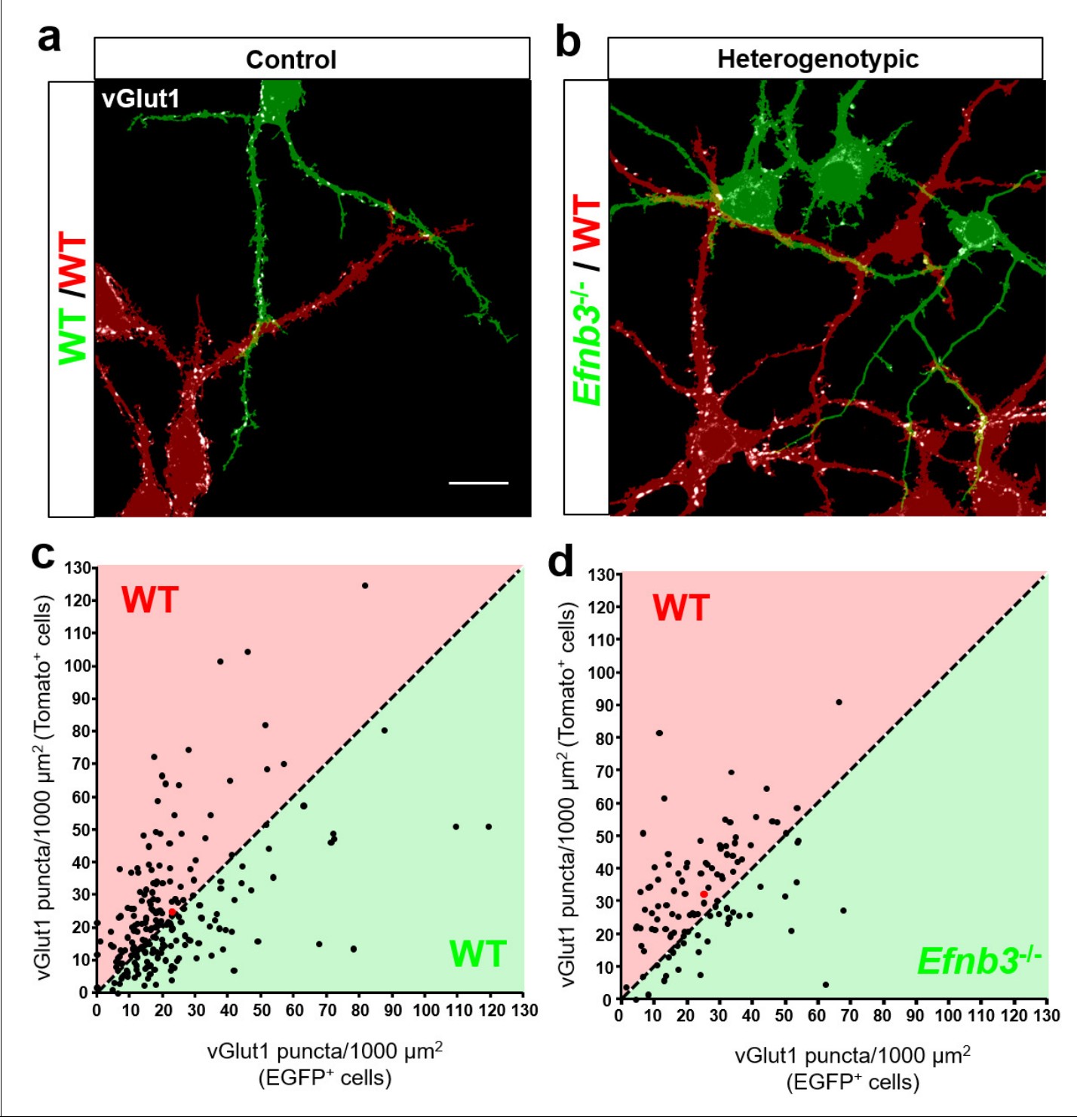

**Figure 8.** Ephrin-B3 regulates local cell-cell differences in synapse density in MADM cultures. (**a and b**) Representative image fields and corresponding masks used for analysis from control (**a**) and heterogenotypic (**b**) DIV11 MADM cultures. Scale bar, 20 μm. (**c and d**) Scatter plots in which each point represents the density of vGlut1+ synapses contacting tdTomato+ (y axis) and EGFP+ (x axis) neurons in a single image field. The red point in each graph represents the average of all data points. The proportion of fields in which tdTomato+ neurons received a higher fraction of local synaptic contacts than EGFP+ neurons is significantly higher in heterogenotypic cultures than control cultures (control, n= 218 fields; heterogenotypic, n=108 fields; p=0.0216, Fisher's exact test).

DOI: https://doi.org/10.7554/eLife.41563.026

The following source data is available for figure 8:

*Figure 8 continued on next page*

*Figure 8 continued*

**Source data 1.** Ephrin-B3 regulates local cell-cell differences in synapse density in MADM cultures.
DOI: https://doi.org/10.7554/eLife.41563.027

synaptic contacts onto WT tdTomato + cells was significantly higher in than in control cultures (74% vs. 58%, p=0.0036, Fisher's exact test) (*Figure 9c,d*). Similar to control MADM cultures, TTX treatment did not significantly change the total density of synapses in heterogenotypic cultures (untreated cultures, n = 3, TTX-treated cultures, n = 3, 26.35 ± 4.43 vs. 19.97 ± 7.28 vGlut1 puncta/1000 $\mu m^2$, t(4)=0.748, p=0.496, two-tailed Student's t-test). Thus, eB3-mediated competition for synaptic inputs persists in the absence of action potential-driven neuronal activity, suggesting that the eB3-EphB competition may function upstream of neuronal activity.

## Discussion

Here, we find that a trans-synaptic organizing molecule, eB3 mediates a cell-cell competition for EphBs that regulates the density of synapses onto cortical neurons. This is supported by three findings: 1) relative differences in eB3 control the distribution of pre-synaptic inputs between isolated pairs of cells without altering total synapse density (*Figure 3*), 2) relative levels of eB3 control spine density in vivo (*Figure 6*), and 3) addition of exogenous EphB2 prevents eB3-dependent competition (*Figure 4*). Further supporting the competition model, neurons that lack eB3 have no synapse density phenotype when cultured with other neurons lacking eB3. However, when challenged with wild-type eB3 expressing neurons, these cells have significantly fewer synapses (*Figure 8*) (*McClelland et al., 2010*). Thus, analogous to activity-dependent competition that is important for critical period plasticity of ocular dominance and orientation selectivity in visual cortex (*Katz and Shatz, 1996*), eB3 appears to regulate a molecular competition for generation or stabilization of synapses at the level of individual neurons.

Cortical neurons in *Efnb3⁻/⁻* mice do not display a synaptic density phenotype, but exhibit reduced synapse density when co-cultured with wild-type neurons (*McClelland et al., 2010*). These findings suggested the surprising possibility that eB3 might direct a competition between adjacent cells to regulate synaptic density. We find that cell-cell differences in eB3 levels in two neuron micro-islands regulate the distribution of synaptic contacts but not total synapse number. Thus, we propose that eB3 functions as a signal that allows cells to compete with one another for pre-synaptic contacts. The absence of a synaptic density phenotype in *Efnb3⁻/⁻* mice is consistent with this model since without eB3 expressed there can be no competition. Also consistent with the model is the finding that in eB3 MADM mice the relative levels of eB3 regulate dendritic spine density in CTIP2 + layer 5 and 6 pyramidal neurons, which express higher levels of eB3 than neighboring SATB2 +/CTIP2 - cells. Our findings suggest that eB3 defines a new class of proteins that regulate an underappreciated aspect of synapse development: the specification of synapse number in relation to neighboring cells.

For what does eB3 compete? The finding that blockade of ephrin-B-EphB interaction rescues synapse density following eB3 knockdown suggests that eB3 likely competes for EphB-expressing axonal contacts and is consistent with the finding that EphB2 can be localized both pre- and post-synaptically in cortex (*Bouvier et al., 2008*). However, it is possible that other Ephs, such as EphA4 may also function as pre-synaptic ligands, or that *cis* interactions between eB3 and EphB2 may be involved (*Antion et al., 2010*; *Takemoto et al., 2002*; *Yokoyama et al., 2001*). Importantly, treatment of control cultures with unclustered EphB2 ectodomain to block ephrin-B-EphB interactions does not alter synapse density, suggesting that it does not directly affect synapse development. In contrast, EphB2 ectodomain fully rescued synapse density following eB3 knockdown (*Figure 4* and *Figure 4—figure supplement 2*). Together, these data suggest that blockade of ephrin-B-EphB interaction does not increase the intrinsic ability of neurons to form synapses following eB3 knockdown, but rather nullifies the competitive advantage of surrounding eB3-expressing cells (*McClelland et al., 2010*). Together with data showing that EphB2 is required for eB3-mediated pre-synaptic differentiation (*McClelland et al., 2010*), the simplest model suggested by our data is that eB3 enables neurons to compete for EphB2-expressing axons.

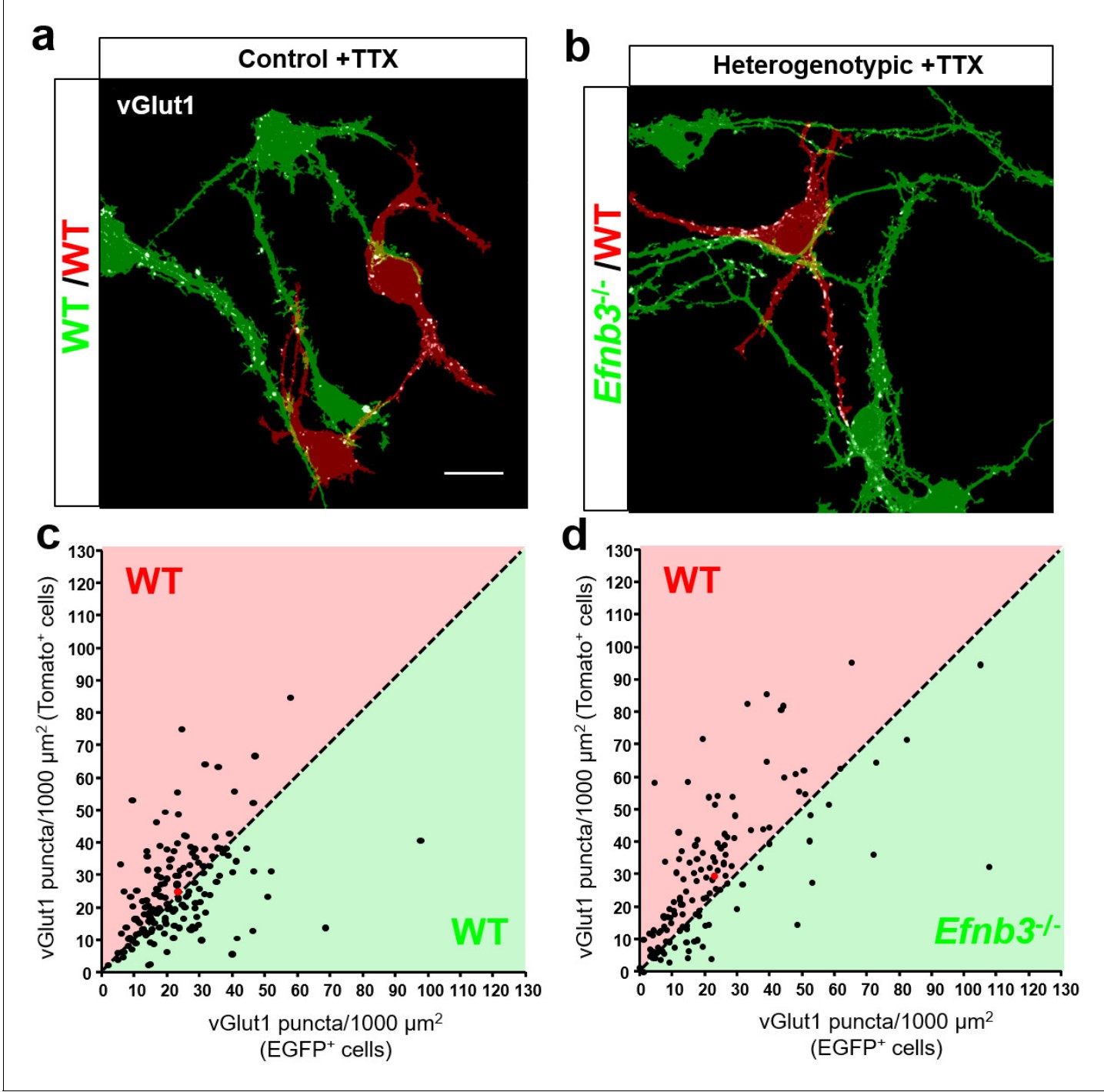

**Figure 9.** Control of synapse number by ephrin-B3 does not require activity. (**a and b**) Representative image fields and corresponding masks used for analysis from control (a) and heterogenotypic (b) DIV11 MADM cultures treated with TTX (1 µm) from DIV3-11. Scale bar, 20 µm. (**c and d**) Scatter plots in which each point represents the density of vGlut1 + synapses contacting tdTomato+ (y axis) and EGFP+ (x axis) neurons in a single image field. The red point in each graph represents the average of all data points. The proportion of fields in which tdTomato + neurons received a higher fraction of local synaptic contacts than EGFP + neurons is significantly higher in heterogenotypic cultures than control cultures (control, n = 166 fields; heterogenotypic, n = 135 fields; p=0.0036, Fisher's exact test).

DOI: https://doi.org/10.7554/eLife.41563.028

The following source data is available for figure 9:

**Source data 1.** Control of synapse number by ephrin-B3 does not require activity.
DOI: https://doi.org/10.7554/eLife.41563.029

It is unlikely that other ephrin-B family members function in a similar fashion to ephrin-B3. Loss of function experiments in cortical neurons have demonstrated that members of the ephrin-B family have non-redundant roles in synapse development. Ephrin-B3 functions post-synaptically, while ephrin-B1 and ephrin-B2 have specific pre-synaptic functions (*McClelland et al., 2010*; *McClelland et al., 2009*). These differences likely arise in part from different subcellular localization of the three ephrin-Bs. In addition, there are important differences in how ephrin-B3 signals that likely mediate its ability to control the competition for excitatory synapses. Ephrin-B3, but not ephrin-B1 or ephrin-B2, contains an Erk-binding D domain that is required for regulation of synapse density (*Hruska et al., 2015*). In addition, eB3 contains a perfect MAPK consensus sequence centered on serine 332 that is absent in ephrin-B1 and ephrin-B2 (*Hruska et al., 2015*). Phosphorylation of serine 332 is dependent on Erk-MAPK signaling and negatively regulates a direct interaction between eB3 and PSD-95. The eB3-PSD-95 interaction stabilizes PSD-95 at synaptic sites and appears to be crucial for eB3 to regulate synapse number (*Hruska et al., 2015*). Thus, within the ephrin-B family, ephrin-B3 has unique signaling motifs and protein interactions that likely enable post-synaptic eB3 to controlling synapse development. Further work will be needed to determine whether these specific signaling mechanisms mediate the molecular competition for synapses.

Though other synaptogenic factors have not been studied in a system where competitive interactions can be isolated, there is evidence that proteins such as Neuroligin-1 (NLGN1) and Trk receptor tyrosine kinases, which have strong knockdown phenotypes but fail to display clear synaptogenesis defects in knockout mice, and may also determine synapse number in a competitive fashion (*English et al., 2012*; *Joo et al., 2014*; *Kwon et al., 2012*; *Varoqueaux et al., 2006*). Despite the lack of a decrease in overall synapse number in NLGN1 knockout (KO) mice, it was found that NLGN1 specifically promotes the formation of thalamocortical synapses by interacting with the astrocyte-secreted protein Hevin, which bridges NLGN1 with pre-synaptic neurexin-1α (*Singh et al., 2016*). NLGN1 KO mice exhibit no change in the overall density of synapses, but a decrease in the number of thalamocortical synapses, suggesting that other sources of synaptic input to cortex compensate for the loss of thalamocortical connections (*Singh et al., 2016*). This raises the intriguing possibility that distinct populations of axons may compete with one another for post-synaptic targets. Determination of whether NLGN1 and Trk receptors mediate a direct competition or regulate the ability of neurons to establish synaptic contacts will require use of a simplified system similar to microislands. Future studies should investigate whether these proteins are also members of a class of trans-synaptic molecules that function to define the distribution of synapses between neighboring neurons.

Early in development, activity-independent molecular cues such as trans-synaptic adhesion molecules establish the initial patterning of synapses, after which neuronal activity plays a key role in refining connectivity while maintaining the proper density of synaptic inputs (*Dalva et al., 2007*; *Flavell et al., 2006*; *Katz and Shatz, 1996*). Our results suggest that the density of synapses onto each neuron can be tuned in relation to its neighbors early in development by competition mediated by eB3. This competition persists in the absence of action potential-driven neuronal activity, suggesting that eB3 may function in an activity-independent pathway. Our findings suggest that while neurons possess activity-dependent mechanisms that regulate the input-output relationship of neurons (*Turrigiano, 2012*), molecular cues can also set the density of excitatory inputs. Ephrin-B3-dependent regulation of synapse density appears to be one such mechanism. It is possible that synapse density and connectivity established by eB3 represents a first step in generating specific patterns of connections which are later refined by activity-dependent competition. Since the eB3-PSD-95 interaction is required for control of synapse number by eB3 (*Hruska et al., 2015*), one possibility is that eB3 may stabilize nascent contacts by recruiting PSD-95 prior to activity-dependent synapse refinement. Interestingly, another trans-synaptic adhesion protein, N-Cadherin, was recently found to be involved in an activity-dependent inter-spine competition that appears to be important for synaptic pruning (*Bian et al., 2015*). Thus, distinct trans-synaptic interactions may be involved in different stages of synapse development and refinement.

In mammalian cortex, neighboring pyramidal neurons exhibit a low connection probability (CP) on average (CP < 0.1). Yet within these sparsely connected circuits, subpopulations of neurons exhibit connection probabilities much higher than what would be expected by chance, with an over-representation of reciprocal connections (*Brown and Hestrin, 2009*; *Song et al., 2005*). These patterns of connections cannot be explained by patterns of axonal growth, as the axon of each neuron

is thought to promiscuously contact the dendrites of all neighboring neurons (*Brown and Hestrin, 2009*; *Kalisman et al., 2005*; *Kasthuri et al., 2015*). Together, these observations suggest that the decision to accept or reject a potential synaptic contact is local, occurring at the site of axo-dendritic contact.

Here, we find that a cell-cell competition mediated by a trans-synaptic organizing protein, eB3, can control the relative distribution of synaptic contacts between neighboring neurons, without altering the overall number of synapses (*Figure 3*). By generating a trans-cellular competition, eB3 may act as a cue that enables neurons to 'interview' potential synaptic partners with appropriate pre-synaptic markers (EphBs). Importantly, expression levels of eB3 and gene dosage of eB3 is correlated with synapse number (*Figure 6*) (*McClelland et al., 2010*). Thus, eB3-dependent competition could allow neurons to fine tune the number of synapses they receive. Consistent with this model, $Efnb3^{+/-}$ neurons within eB3 MADM mice exhibited spine densities in between that of WT and $Efnb3^{-/-}$ neurons in the same mice. Interestingly, the difference between WT and $Efnb3^{+/-}$ neurons reached significance, while the difference between $Efnb3^{+/-}$ neurons and $Efnb3^{-/-}$ neurons did not. Given the large effects seen in heterogenotypic cultures or following eB3 knockdown (*McClelland et al., 2010*), these findings are somewhat surprising and, suggest either that there may be some compensation in the MADM mice, or that the presence of many unlabeled $Efnb3^{+/-}$ neurons in MADM eB3 mice reduces the impact of the loss of eB3. Regardless, variations in the level of eB3 appear to regulate the density of excitatory synaptic connections cortical neurons receive.

Ephrin-B3-EphB2-mediated competition appears ideally suited to generate a key feature of cortical connectivity: highly interconnected sub-populations of cells within an overall sparsely connected network, a feature that may be crucial for optimizing the storage capacity of cortical networks (*Brunel, 2016*). One example of this type of connectivity is in visual cortex, where the connection probability between neighboring cells differs between subpopulations of layer five neurons (*Brown and Hestrin, 2009*). Subcortically projecting neurons (corticotectal and corticostriatal) receive a high density of inputs from callosally projecting and corticostriatal neurons, respectively. In contrast, the local inputs onto neighboring callosally projecting layer 5 cells are relatively sparse. In light of the finding that CTIP2+, subcortically projecting neurons express higher levels of *Efnb3* mRNA than SATB2+/ CTIP2 -, callosally projecting neurons, one function of eB3-dependent competition may be to increase the relative density of specific types of inputs to subcortically projecting cells compared to neighboring callosally projecting cells. Consistent with previous work (*McClelland et al., 2010*), we also observed considerable variability in eB3 expression between individual CTIP2 + cells, suggesting that eB3 may serve to finely tune the connectivity of individual neurons within this population. More work will be needed to determine whether loss of eB3 impacts these specific types of circuit organization.

# Materials and methods

## Key resources table

| Reagent type (species) or resource | Designation | Source or reference | Identifiers | Additional information |
|---|---|---|---|---|
| Genetic reagent (*Mus musculus*) | ephrin-B3 knockout (and WT littermates) | *Yokoyama et al., 2001*, *Hruska et al., 2015*, *Antion et al., 2010*. | RRID: MGI:3026744 | |
| Genetic reagent (*M. musculus*) | MADM-11 TG, eB3+/MADM-11 TG, eB3- (founder line) | This paper. | | Progenitor: MADM-11 TG/TG (*Hippenmeyer et al., 2010*), RRID: IMSR_ JAX:013749 |
| Genetic reagent (*M. musculus*) | MADM-11 GT/GT; Emx1-Cre | *Hippenmeyer et al., 2010*, *Beattie et al., 2017*. | | Progenitor: IMSR_JAX:005628 |

*Continued on next page*

*Continued*

| Reagent type (species) or resource | Designation | Source or reference | Identifiers | Additional information |
|---|---|---|---|---|
| Genetic reagent (*M. musculus*) | MADM-11 GT/GT; Nestin-spCre | *Hippenmeyer et al., 2010*. | | |
| Strain, strain background (*Rattus norvegicus domesticus*) | Long Evans Rat | Charles River | Strain code: 006: RRID: RGD_2308852 | |
| Aantibody | Rabbit polyclonal anti-GFP | Thermo Fisher Scientific | A6455, RRID: AB_221570 | (1:2500) |
| Aantibody | Mouse monoclonal anti-PSD-95 | NeuroMab (clone 28/43) | Cat. # 75–028, RRID:AB_ 2292909 | (1:500) |
| Antibody | Guinea pig polyclonal anti-vGlut1 | Millipore | Cat. #AB5905, RRID:AB_ 2301751 | (1:5000) |
| Antibody | Rabbit polyclonal anti-RFP | Rockland antibodies | Cat. # 600-401-379, RRID:AB_ 2209751 | (1:500) |
| Antibody | Chicken polyclonal anti-GFP | Abcam | Cat. # ab13970, RRID:AB_ 300798 | (1:500) |
| Antibody | Rat monoclonal anti-CTIP2 | Abcam | Cat. # ab18465, RRID:AB_ 10015215 | (1:500) |
| Antibody | Mouse monoclonal anti-SATB2 | Abcam | Cat. # ab51502, RRID:AB_ 882455 | (1:500) |
| Antibody | Donkey polyclonal anti-rabbit Dylight 488 | Abcam | Cat. # ab96919, RRID:AB_ 10679362 | (1:500) |
| Antibody | Donkey polyclonal anti-chicken Dylight 488 | Jackson Immunoresearch | Cat. # ab96947, RRID:AB_ 10681017 | (1:500) |
| Antibody | Donkey polyclonal anti-rabbit Alexafluor 594 | Jackson Immunoresearch | Cat. # 711-585-152, RRID:AB_ 2340621 | (1:500) |
| Antibody | Donkey polyclonal anti-rat 647 | Jackson Immunoresearch | Cat. # 712-605-153, RRID:AB_ 2340694 | (1:500) |
| Antibody | Goat polyclonal anti-mouse ATTO 425 | Rockland antibodies | Cat. # 611-151-122, RRID:AB_ 10893217 | (1:250) |
| Antibody | Donkey polyclonal anti-guinea pig Dylight 649 | Jackson Immunoresearch | Cat. # 706-605-148, RRID:AB_ 2340476 | (1:500) |

*Continued on next page*

*Continued*

| Reagent type (species) or resource | Designation | Source or reference | Identifiers | Additional information |
|---|---|---|---|---|
| Recombinant DNA reagent | pFUGW-tdTomato_H1-eB3.2 (plasmid) | *Hruska et al., 2015*. | | Progenitor: pFUGW (RRID:Addgene_14883); pSuper. |
| Recombinant DNA reagent | pFUGW-tdTomato_H1-pSuper (plasmid) | *Hruska et al., 2015*. | | Progenitor: pFUGW (RRID:Addgene_14883), pSuper. |
| Recombinant DNA reagent | pSuper (plasmid) | *Hruska et al., 2015, McClelland et al., 2010*. | | |
| Recombinant DNA reagent | eB3.2 shRNA (plasmid) | *Hruska et al., 2015, McClelland et al., 2010*. | | pSuper |
| Recombinant DNA reagent | pFUGW (plasmid) | *Hruska et al., 2015, McClelland et al., 2010*. | RRID:Addgene_14883 | |
| Recombinant DNA reagent | pLL3.7 (plasmid) | Addgene | RRID:Addgene_11795 | |
| Recombinant DNA reagent | ephrin-B3 shRNA lentivirus | Penn Vector Core (University of Pennsylvania) | | Progenitor: pLL3.7 vector (RRID:Addgene_11795) |
| Peptide, recombinant protein | Recombinant human EphB2 Fc chimera | R and D systems | Cat. # 5189-B2 | |
| Peptide, recombinant protein | Recombinant Fc control fragment | R and D systems | 110-HG-100 | |
| Commercial assay or kit | RNAscope fluorescent multiplex kit | Advanced Cell Diagnostics | 320850 | |
| Commercial assay or kit | RNAscope probe- Mm-Bcl11b (CTIP2) | Advanced Cell Diagnostics | Cat. # 413271-C3 | |
| Commercial assay or kit | RNAscope probe- Mm-Satb2-C2 | Advanced Cell Diagnostics | Cat. # 413261-C2 | |
| Commercial assay or kit | RNAscope probe- Mm-Efnb3 | Advanced Cell Diagnostics | Cat. # 526771 | |
| Chemical compound, drug | Tetrodotoxin (TTX) | Tocris | Cat. # 1078 | |
| Software, algorithm | NIH ImageJ | *McClelland et al., 2009, Hruska et al., 2015*. | RRID: SCR_003070 | |

## Constructs used for transfection

We used GATEWAY technology (Fisher Scientific) to generate constructs that lead to simultaneous expression of PSD-95-GFP, tdTomato and either ephrin-B3 shRNA or control pSuper cassette (pFUGW-tdTomato-eB3.2 and pFUGW-tdTomato-pSuper) for use in microisland cultures. The promoters used are from human ubiquitin (hUb), synapsin (Syn), and H1, respectively. To do this, the existing pFUGW vector was converted into a GATEWAY destination vector using the Gateway vector conversion system kit (Fisher Scientific). First, we removed GFP (EGFP) downstream of the hUb promoter from the original pFUGW by cutting with XbaI (New England Biolabs) and subsequently

blunted the sites with Klenow fragment (New England Biolabs). Then in place of GFP we cloned Reading Frame A cassette (RfA, Fisher Scientific) that contains 5' attR1 and 3' attR2 sites, with chloramphenicol and ccdB genes, thus rendering pFUGW Gateway compatible. Finally, to generate the expression plasmid containing PSD-95-GFP, tdTomato and ephrin-B3 shRNA with their respective promoters, we performed 4-fragment Gateway LR recombination by combining the pFUGW-Destination vector and four pDONR 221 entry plasmids containing following fragments: 1) 5' attL1_PSD-95-GFP_attR5 3'; 2) 5' attL5_BGH_PA_attL4 3'; 3) 5' attR4_SynPro_tdTomato_SV40PA_attR3r 3' and 4) 5' attL3_H1_ephrin-B3shRNA_attL2 3' or 5' attL3_pSuper cassette_attL2 3'. The resulting pFUGW expression plasmids contained PSD-95-GFP under control of the hUb promoter, tdTomato under control of the synapsin promoter and ephrin-B3 shRNA or pSuper cassette under control of the H1 promoter. For complex neuronal culture experiments, neurons were transfected with pFUGW to express GFP together with eB3.2 shRNA or pSuper control vectors (*Hruska et al., 2015* ).

## Generation of ephrin-B3 MADM mice

MADM-11 mice containing the TG and GT knock-in cassettes to the *Hipp11* locus on chromosome 11 (*Hippenmeyer et al., 2010*) were obtained from Dr. Simon Hippenmeyer. *MADM-11$^{TG/TG}$* mice were crossed to mice heterozygous for an ephrin-B3 null allele (*Yokoyama et al., 2001*) to obtain *MADM-11 TG, Efnb3$^+$+ MADM-11 TG, Efnb3$^-$* (founder line) mice. *MADM-11$^{GT/GT}$* mice were crossed to either an *Emx1-Cre* line, or an *spNestin-Cre* (*Nestin-CreER$^{T2}$*) line where CRE recombinase is active in a sparse, random subset of neural progenitor cells in the absence of tamoxifen (*Hippenmeyer et al., 2010*), to obtain *MADM-11$^{GT/GT}$; Emx1-Cre* and *MADM-11$^{GT/GT}$; Nestin-spCre* lines, respectively. These lines were crossed to the founder line, yielding *MADM-11$^{GT/TG, Efnb3+/-}$; Emx1-Cre$^{+/-}$* (eB3 MADM; Emx1-Cre) or *MADM-11$^{GT/TG,Efnb3+/-}$; Nestin-spCre$^{+/-}$* (eB3 MADM; Nestin-spCre) mice.

## Complex neuronal cultures

E17-18 Long Evans rat cortical neurons were dissociated and cultured as described previously (*McClelland et al., 2010*). Briefly, neurons were cultured in 24 well plates at a density of 150,000 cells/well in Neurobasal media (Fisher Scientific) supplemented with B27 supplement (Fisher Scientific), glutamine (Sigma), and penicillin/streptomycin (Sigma). Coverslips (12 mm Bellco Glass) were coated with poly-D-lysine and laminin (BD Biosciences). For transfections at DIV0, cells were dissociated in OptiMEM (Fisher Scientific) and transfected using Lipofectamine (Fisher Scientific) immediately before plating.

## MADM neuronal cultures

Neurons were dissociated from P0-1 Emx1-Cre control and eB3 MADM mice. Labeled neurons were then isolated by FACS sorting, and EGFP + and tdTomato + cells were plated at a 1:1 ratio in 48 well plates at a total density of approximately 30,000 cells/well. Neurons were cultured in Neurobasal media (Fisher Scientific) supplemented with B27 supplement (Fisher Scientific), glutamine (Sigma), and penicillin/streptomycin (Sigma). Coverslips (5 mm Bellco Glass) were coated with poly-D-lysine and laminin (BD Biosciences). From DIV3-11, neurons were treated with culture media containing TTX (1 µm final concentration) or nothing added. Cultures were also treated every 48 hr with mitosis inhibitors uridine (10 µm) and 5-fluoro-2-deoxyuridine (0.1 µm) to prevent any astrocytes present from dividing.

## Microisland cultures

Microislands were generated from E17-18 dissociated rat cortical neurons using a protocol modified from *Allen (2006)*. Coverslips (12 mm Bellco Glass) were placed in 24-well plates, then 10–30 µl of a 0.15% agarose solution was applied to each coverslip and allowed to air dry. 3 drops of heated paraffin wax (Aldrich) were applied to the edges of each coverslip, and plates were stored at 4 °C until further use. On the day neurons were plated, an atomizer was used to spray droplets of poly-D-lysine/laminin solution into the pre-prepared 24 well plates described above. After air-drying, the plates were washed once with distilled water. 500 ul of supplemented Neurobasal media (see above) was added to each well and the plates were placed in a humidified incubator maintained at 5% CO2°C and 37°C. Prior to plating, neurons were transfected by electroporation using an Amaxa

Nucleofector (Lonza Walkerrsville). 5 million neurons per transfection condition were suspended in 100 ul of P3 primary cell solution (Lonza cat. no. PBP3-02250) and transfected with 3 µg of the appropriate DNA construct. Immediately after transfection, 900 µl of pre-warmed, supplemented Neurobasal medium was added to the transfected cells, which were subsequently plated in the pre-warmed 24 well plates described above at a density of 15,000–40,000 cells/well. On the same day as transfection, separate 'feeder' layers of neurons were prepared by coating 24 well plates (without coverslips) with poly-D-lysine/Laminin solution and plating 150,000 cells/well. The following day, the coverslips containing the transfected neurons were placed onto the feeder neurons (wax droplets facing down). Cultures were maintained in this manner in a humidified incubator maintained at 5% CO2°C and 37°C.

## Lentiviral transduction

ephrin-B3 shRNA target sequence was cloned into lentilox pLL3.7 GFP packaging plasmid and the high titer ($10^{10}$) lentivirus was produced by Penn Vector Core at the University of Pennsylvania. Neurons were transduced with control or eB3 shRNA lentivirus immediately after dissociation by adding 1–5 µl of high titer particles to the neuronal suspension containing 5–10 million neurons. After incubation at 37°C for 2 hr, neurons were plated on coverslips containing PDL/laminin spotted microdots and placed into 37°C/5% $CO_2$ incubator for the indicated amount of time before synapse density analysis was performed.

## Unclustered EphB2 ectodomain treatment

Cultured neurons in 24 well plates were treated with unclustered EphB2-Fc or control Fc fragment (R and D Biosystems) once every 48 hr starting from DIV3 to DIV10 (complex cultures) or DIV21 (micro-island cultures). To test the dose-dependency of EphB2-Fc treatment, separate treatment conditions of 2.5 ng/well (5 ng/ml), 25 ng/well (50 ng/ml), and 250 ng/well (500 ng/ml) were used. To determine whether neurons expressing ephrin-B3 shRNA bound less EphB2-Fc, DIV10 control and ephrin-B3 shRNA-expressing neurons were treated with 500 ng/ml with unclustered EphB2-Fc for 45 min, then fixed in 4% PFA with 2% sucrose for immunocytochemistry.

## Immunoprecipitation and western blots following EphB2 ectodomain treatment

DIV8 cortical neurons were treated with control Fc fragment, 500 ng/ml pre-clustered EphB2-ectodomain, or 500 ng/ml unclustered EphB2-ectodomain for 45 min. Neurons were then lysed in 500 µl of lysis buffer (50 mM Tris, 140 mM NaCl, 0.5% NP40, pH7.4) containing general protease inhibitor cocktail (Sigma) and 1 mM PMSF and phosphatase inhibitors (2 mM EGTA (Sigma), 50 mM NaF (Sigma), 1 mM $Na_3VO_4$ (Sigma), 10 mM $Na_4O_7P_2.10H_2O$ (Sigma) and 0.1 mM $H_{24}Mo_7N_6O_{24}.4H_2H$ (Sigma). Insoluble material was removed by centrifuging at 10,000 g for 20 min at 4°C and the remaining supernatant was immunoprecipitated with a rabbit anti-ephrin-B3 antibody (Thermo) for 4 hr at 4°C. Then 20 µl of protein G beads (blocked in 1% BSA in lysis buffer) was added to the supernatant and incubated by head over tail rotation for an additional hour at 4°C. Beads were washed three times with lysis buffer and once with Tris-buffered saline containing 10 mM NaF, 1 mM $Na_3VO_4$ with protease inhibitor cocktail (Sigma) and 1 mM PMSF. Proteins were subsequently eluted from the beads by boiling in 4x SDS sample buffer for 5 min. Eluted proteins were separated on SDS-PAGE gels, transferred to PVDF membrane (Millipore), blocked with 1% BSA, labeled with anti ephrin-B3 (Thermo) and anti phospho-ephrin-B (Cell Signaling) antibodies and HRP-conjugated secondary antibodies, and visualized with Western Lightning Plus-ECL (PerkinElmer, Waltham, MA).

## Immunocytotochemistry

Cultured neurons were fixed in paraformaldehyde (PFA) solution containing 4% PFA and 2% sucrose for 8 min at room temperature, washed 3X in PBS, then blocked for 1 hr at room temperature in a solution containing 1% ovalbumin (Sigma)/0.2% cold-water fish scale gelatin (Sigma) and 0.1% saponin. Primary antibody incubations were performed at 4 °C overnight. The antibodies used were as follows: rabbit anti-GFP (1:2500, Fisher Scientific), mouse anti-PSD-95 (1:500, NeuroMab), guinea pig anti-vGlut1 (1:5000, Millipore), rabbit anti-RFP (1:500, Rockland antibodies), and chicken anti-GFP (1:500, Abcam). Secondary antibody incubations were performed at room temperature for 45

min. The secondary antibodies were used at a concentration of 1:250–500 and were as follows: donkey anti-rabbit Dylight 488 (Abcam), donkey anti-guinea pig Dylight 649 (Jackson Immunoresearch), and goat anti-mouse ATTO 425 (Rockland Immunochemicals). Coverslips were mounted with Aquamount aqueous media (Lerner Laboratories).

## Immunohistochemistry

In order to achieve rapid fixation, 300 μm acute brain slices from postnatal day (P)18–21 Nestin-spCre control and eB3 MADM-11 mice were cut, maintained for two hours in artificial cerebrospinal fluid (ACSF), then fixed overnight at 4 ˚C by rapid submersion in freshly prepared 4% PFA. Brain sections were then washed three times in 1X PBS and submerged in 30% Sucrose/PBS overnight. The sections were then re-sectioned at 40 μm. For immunostaining, sections were permeabilized for 25 min in 0.5% Triton X-100, then blocked overnight at 4 ˚C in blocking solution containing 10% FBS, 1% BSA, and 0.2% Triton X-100. The following day, sections were incubated with primary antibodies in blocking solution overnight at 4 ˚C. The following day, sections were incubated for 2 hr at room temperature with the appropriate secondary antibodies. The primary antibodies used were as follows: chicken anti-GFP (1:500, Abcam), rabbit anti-RFP (1:500, Rockland antibodies), rat anti-CTIP2 (1:500, Abcam), mouse anti-Satb2 (1:500, Abcam). The secondary antibodies used were as follows: donkey anti-chicken dylight 488 (1:500, Abcam), donkey anti-rabbit Alexafluor 594 (1:500, Jackson Immunoresearch), donkey anti-rat Alexafluor 647 (1:500, Jackson Immunoresearch), and goat anti-mouse ATTO 425 (1:250, Rockland antibodies).

## RNAscope ISH

P28 WT and *Efnb3*[−/−] mice were trans-cardially with PBS followed by 4% PFA. Brains were post-fixed for 24 hr at 4˚ C, then washed 3X with PBS and submerged in 30% sucrose/PBS for 48 hr at 4˚C. Brains were then frozen in OCT medium, cryosectioned at 20 μm, and mounted on Superfrost slides (Fisher Scientific). Slides were stored at −80 ˚C until use, then baked at 50 ˚C for one hour prior to initiating the RNAscope protocol. Pre-treatment of tissue sections and RNAscope ISH was carried out according to manufacturer protocols (Advanced Cell Diagnostics, Inc) using the RNAscope fluorescent multiplex kit (Cat No. 320850) with CTIP2 (Cat No. 413271-C3), SATB2 (Cat No. 413261-C2) and eB3 (Cat No. 526771) probes.

## Imaging and analysis

### Complex neuronal cultures and microislands

Images were aquired on a Leica TCS SP5 confocal scanning microscope (Leica Microsystems, Mannheim, Germany). Stacks of 6–8 images taken at 0.3 μm intervals were acquired with a 63X oil immersion objective (Leica) with a 1.7X-1.9X zoom. Maximum intensity projections of the image stacks were analyzed to calculate synapse or puncta density using custom ImageJ macros as described previously (*McClelland et al., 2010*). Briefly, each channel for each image was thresholded and converted to a binary image, and vGlut1, and PSD-95 puncta were defined as 0.5–7.5 μm of continuous pixels. For unclustered EphB2 ectodomain control experiments, puncta were defined as having at least one pixel overlap with GFP-labeled dendrites. Synapse or puncta density was calculated as the density of colocalized puncta (with > 1 pixel overlap) from a region of at least 50 μm of dendrite per cell. In two-neuron microislands, the number of colocalized PSD-95-GFP/vGlut1 puncta was used to calculate synapse density in transfected neurons. For untransfected neurons, synapse density was calculated by subtracting the number of colocalized PSD-95-GFP/vGlut1 puncta from the number of PSD-95/vGlut1 puncta.

### MADM cultures

MADM cultures were imaged on a Leica TCS SP5 confocal scanning microscope (Leica Microsystems, Manheim, Germany). The automated imaging feature in Leica application suite was used to sequentially image the entirety of each coverslip with a 63X oil immersion objective (Leica) and 1.7X zoom. Image fields containing EGFP + and tdTomato + cells were then selected for analysis. We confirmed that the numbers of tdTomato + and EGFP + neurons in each culture did not significantly differ by selecting 6–10 random image fields form each experiment and counting the number of EGFP + and tdTomato + cell bodies in each image. The numbers of EGFP + and tdTomato + cells did not

significantly differ in control (n = 26 fields, 2.46 ± 0.347 red vs. 2.231 ± 0.29 green cells, t(50)= 0.51, p=0.6123, two-tailed Student's t-test) or heterogenotypic (n = 25 fields, 2.44 ± 0.252 red vs. 3.16 ± 0.496 green, t(48)= 1.295, p=0.2016, two-tailed Student's t-test) cultures. Fields with exceedingly high cell density were excluded due to the difficulty of accurately assigning synaptic contacts to cells. In addition, fields containing cells with non-neuronal morphology were excluded. Analysis was carried out using custom ImageJ macros. Briefly, maximum intensity projections for each channel for each image were generated, thresholded and converted to binary images. Axons were removed from tdTomato and EGFP channels using the particle analysis tool in ImageJ. vGlut1 puncta were defined as 0.5–7.5 μm of continuous pixels. An ImageJ macro was used to assess the overlap of each vGlut1 puncta with tdTomato and EGFP channels, assigning each puncta to the channel that contained the largest area of overlap with the vGlut1 puncta. Puncta that overlapped equally with each channel were categorized as 'unknown', and manually assigned when possible. Synapse density was then calculated on a per image basis as density per unit area of vGlut1 puncta assigned to tdTomato + and EGFP + cells.

## Dendritic spine imaging and quantification

MADM brain sections were imaged on a Leica TCS SP5 confocal scanning microscope (Leica Microsystems, Mannheim, Germany). Stacks of 5–15 images taken at 0.6–2 μm intervals were acquired with a 63X oil immersion objective (Leica) with 1.7–2.4X zoom. For each cell, dendritic spines were counted on a segment of apical dendrite at least 40 μm in length. Dendrite width was measured by creating a mask of each dendrite, manually erasing dendritic protrusions from the image, then calculating the average width of the dendrite by dividing the total area of the dendrite by its length.

## RNAscope ISH imaging and quantification

WT and $Efnb3^{-/-}$ brain sections were imaged on a Leica TCS SP8 confocal scanning microscope (Leica Microsystems, Mannheim, Germany). 10–18 μm stacks (approximately the width of a single nucleus) were acquired at 2 μm intervals with a 63X oil immersion objective and 1.7X zoom. To avoid experimenter bias, fields of view were chosen by visualizing CTIP2 and DAPI signals to identify subgranular layers of cortex, without looking at the eB3 channel. Images were analyzed using custom ImageJ macros. Briefly, maximum projections of each channel were thresholded after applying gaussian blur and background subtraction (same threshold for all RNAscope probes), and nuclei were automatically identified and selected as ROIs. RNAscope puncta in the size range of 4–100 pixels were selected for further analysis. Each ROI corresponding to a nucleus was automatically scored as 'positive' for CTIP2 and/or SATB2 if it contained at least 10 puncta. Ephrin-B3 probe puncta within each of these ROIs were then automatically counted.

## Statistical analyses

Data plotted on bar graphs are expressed as the mean ± SEM. Sample sizes were not formally calculated but are consistent with the standards in field. Statistical differences were determined using one-way ANOVA followed by Tukey's HSD post-hoc test, one sample t-test, Fisher's exact test, or Student's t-test as indicated. P values of less than 0.05 were considered statistically significant, and data were pooled from a minimum of three independent experiments unless otherwise indicated. Statistical tests were done on a per cell basis unless otherwise noted. Experimental groups were defined by treatment/transfection condition or genotype. No outlying data points were removed from analysis. Analysis and data aquisition was done with the experimenter blinded to condition with the exception of MADM culture and RNAscope experiments, for which the analysis was automated. For each experiment, cell morphology was examined and cells that exhibited signs of poor health were excluded.

# Additional information

## Funding

| Funder | Grant reference number | Author |
|---|---|---|
| Howard Hughes Medical Institute | | Liqun Luo |
| National Institute of Neurological Disorders and Stroke | RO1NS050835 | Liqun Luo |
| National Institute on Drug Abuse | R01DA022727 | Matthew B Dalva |
| National Institute of Neurological Disorders and Stroke | R01NS106906 | Matthew B Dalva |
| Farber Family Fund | | Matthew B Dalva |

The funders had no role in study design, data collection and interpretation, or the decision to submit the work for publication.

## Author contributions

Nathan T Henderson, Conceptualization, Software development, Formal analysis, Investigation, Writing First Draft Writing—review and editing; Sylvain J Le Marchand, Martin Hruska, Conceptualization, Formal analysis, Investigation, Writing—review and editing; Simon Hippenmeyer, Liqun Luo, Resources, Writing—review and editing; Matthew B Dalva, Conceptualization, Supervising, Fundraising, Writing—review and editing

## Author ORCIDs

Nathan T Henderson http://orcid.org/0000-0002-9285-6687
Sylvain J Le Marchand https://orcid.org/0000-0003-1831-9725
Martin Hruska http://orcid.org/0000-0003-3186-770X
Simon Hippenmeyer http://orcid.org/0000-0003-2279-1061
Liqun Luo http://orcid.org/0000-0001-5467-9264
Matthew B Dalva http://orcid.org/0000-0002-7737-8787

## Ethics

Animal experimentation: All animal studies were approved by the Institutional Animal Care and Use Committee guidelines at Thomas Jefferson University in accordance with US National Institutes of Health guidelines (01289 and 01286). All surgery was performed under anesthesia, and every effort was made to minimize suffering.

## Decision letter and Author response

Decision letter https://doi.org/10.7554/eLife.41563.031
Author response https://doi.org/10.7554/eLife.41563.032

# Additional files

## Supplementary files

• Transparent reporting form
DOI: https://doi.org/10.7554/eLife.41563.030

## Data availability

All data generated are included in the figures and text of this manuscript.

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
