## [Decision Letter]

Thank you for submitting your article "Ephrin-B3 controls excitatory synapse density through cell-cell competition for EphBs" for consideration by *eLife*. Your article has been reviewed by three peer reviewers, one of whom is a member of our Board of Reviewing Editors, and the evaluation has been overseen by Gary Westbrook as the Senior Editor. The reviewers have opted to remain anonymous. The reviewers have discussed the reviews with one another and the Reviewing Editor has drafted this decision to help you prepare a revised submission.

Summary:

This study by Henderson and colleagues investigates the role of intercellular competition in synapse formation. The authors use cell culture and in vivo approaches to address the consequences of reducing ephrin-B3 levels on synapse density. They find that neurons with reduced ephrin-B3 levels are less competitive in forming synapses in a microisland assay. Treatment with excess unclustered EphB2 abolishes the competitive disadvantage of ephrin-B3 shRNA-expressing neurons, suggesting that the cells compete for presynaptic EphB2. To test this model in vivo, the authors use the MADM approach to analyze spine density in sparsely labeled WT or ephrin-B3 KO neurons surrounded by ephrin-B3 heterozygous cells. They find that WT neurons have higher spine density than KO neurons. Finally, the authors show that neuronal activity is not required for ephrin-B3 mediated competition for synapses.

This is a well-written paper with a clear presentation. The experimental approach is elegant and the topic of cell-cell competition in synapse formation is interesting and not well understood. The main limitation is that compared to a previous study from the same group (McClelland PNAS 2008), the conceptual advance resulting from the current work is limited. The McClelland study already showed that ephrin-B3 mediated competition regulates synapse density independent of neuronal activity, albeit in a less elegant way.

Essential revisions:

1) Based on the competition experiments with excess soluble EphB2, the authors conclude that neurons compete for presynaptic EphB2. They cite Bouvier et al., 2008, in demonstrating that EphB2 is presynaptic, but this study actually shows a mostly dendritic localization of EphB2 in hippocampus and cortex. To make the point that neurons compete for presynaptic EphB2, the authors should demonstrate this directly by knocking down/out EphB2 in axons. If their model is correct, removing EphB2 presynaptically would abolish the competitive disadvantage of ephrin-B3 shRNA neurons. This can be done in the microisland assay with an shRNA for EphB2 expressed in both neurons.

2) To support their model, the authors should provide evidence for varying expression levels of ephrin-B3 (relative to another constantly expressed marker) in CTIP2+ cortical layer 5 and 6 neurons in cortical sections using fluorescent in situ hybridization or immunohistochemistry. Do CTIP2-/SATB2+ neurons not express ephrin-B3 because their dendrites do not seem sensitive to ephrin-B3 levels?

3) In their previous study (McClelland et al.) it was shown that spine density is strongly (60%) reduced in cortical neurons in the ephrin-B3 global KO (no competition between neurons; synapse number in cortical sections based on antibody staining however appeared unchanged). Another study showed that the density of shaft synapses is decreased in ephrin-B3 KO hippocampus using EM (Aoto JNeurosci 2007). This data does not seem entirely compatible with the current study, in which spine density is reduced in MADM mosaic KO neurons compared to WT neurons. In the MADM mouse, WT and KO neurons compete with a majority of ephrin-B3 heterozygous neurons. What is the spine density in heterozygous neurons and do WT and KO neurons differ significantly from these? In Figure 5B, is spine density in KO neurons significantly different from control MADM neurons? This does not seem to be the case in the graph and should be explained.

---

## [Author Response]

Essential revisions:1) Based on the competition experiments with excess soluble EphB2, the authors conclude that neurons compete for presynaptic EphB2. They cite Bouvier et al., 2008, in demonstrating that EphB2 is presynaptic, but this study actually shows a mostly dendritic localization of EphB2 in hippocampus and cortex. To make the point that neurons compete for presynaptic EphB2, the authors should demonstrate this directly by knocking down/out EphB2 in axons. If their model is correct, removing EphB2 presynaptically would abolish the competitive disadvantage of ephrin-B3 shRNA neurons. This can be done in the microisland assay with an shRNA for EphB2 expressed in both neurons.

The first essential revision requested by the reviewers entails providing additional evidence that EphB2 is the pre-synaptic ligand that ephrin-B3 competes for. We feel that this critique may stem in part from a lack of clarity on our part in describing the published evidence that EphB2 is the pre-synaptic ligand for ephrin-B3. In addition to the Bouvier et al. paper mentioned by the reviewers, our lab has previously shown that ephrin-B3 requires pre-synaptic EphB2 to induce pre-synaptic differentiation in a heterologous cell co-culture assay (McClelland et al., 2010). The reviewers also requested experiment where we knock down EphB2 in both cells within a competitive microisland, with the expectation that this would block competition if EphB2 is indeed the pre-synaptic ligand. We have considered this experiment ourselves, but feel that the results would be very difficult to interpret in light of the fact that both pre- and post-synaptic EphB2 would be eliminated in the system. Therefore, we asked for and were given permission by the editor to respond to this criticism with changes to the text. These change are found in the subsection “Ephrin-B3 competes for EphBs”.

2) To support their model, the authors should provide evidence for varying expression levels of ephrin-B3 (relative to another constantly expressed marker) in CTIP2+ cortical layer 5 and 6 neurons in cortical sections using fluorescent in situ hybridization or immunohistochemistry. Do CTIP2-/SATB2+ neurons not express ephrin-B3 because their dendrites do not seem sensitive to ephrin-B3 levels?

We agree with the reviewers that information about the expression of ephrin-B3 in relation to CTIP2 and Satb2 would greatly strengthen the manuscript. We are therefore conducted RNAscope ISH experiments to simultaneously visualize and quantify the expression of ephrin-B3 mRNA (*Efnb3*), CTIP2 mRNA (*Bcl11b*) and SATB2 mRNA (*Satb2*) in brain sections. Consistent with our previous data, we found that CTIP2+ and CTIP2+/SATB2+ neurons in layer 5 and 6 neurons express significantly higher levels *of Efnb3* than SATB2+/CTIP2- cells (Figure 5, subsection “Relative levels of ephrin-B3 determine spine density in vivo”). The levels of eB3 probe signal in SATB2+/CTIP2- cells were not significantly different than CTIP2+, CTIP2+/SATB2+, or SATB2+/CTIP2- cells in ephrin-B3 knockout cortex, suggesting that SATB2+ express little or no eB3. Moreover, the amount of eB3 RNA varied between different CTIP2+ neurons. Thus, CTIP2+ neurons express more ephrin-B3 than SATB2+ neurons and the level of expression varies from CTIP2+ neuron to CTIP2+ neuron. These data are show in Figure 5 and Figure 5—figure supplement 2.

3) In their previous study (McClelland et al.) it was shown that spine density is strongly (60%) reduced in cortical neurons in the ephrin-B3 global KO (no competition between neurons; synapse number in cortical sections based on antibody staining however appeared unchanged). Another study showed that the density of shaft synapses is decreased in ephrin-B3 KO hippocampus using EM (Aoto JNeurosci 2007). This data does not seem entirely compatible with the current study, in which spine density is reduced in MADM mosaic KO neurons compared to WT neurons. In the MADM mouse, WT and KO neurons compete with a majority of ephrin-B3 heterozygous neurons. What is the spine density in heterozygous neurons and do WT and KO neurons differ significantly from these? In Figure 5B, is spine density in KO neurons significantly different from control MADM neurons? This does not seem to be the case in the graph and should be explained.

We agree that the phenotype of the ephrin-B3 heterozygous neurons is interesting and relevant to our study. Based on our model, we would predict that within the ephrin-B3 MADM mice, these neurons would exhibit spine densities in between those of wild-type and ephrin-B3 knockout neurons. We have conducted to further experiments to quantify spine density on these ephrin-B3 heterozygous neurons in brain sections from MADM mice, which are now included in Figure 6. As expected, we found that, on average, spine density on the heterozygous neurons was in between that of wild-type and ephrin-B3 KO neurons.

This new data also sheds light on the reviewers’ second concern, namely the lack of a significant difference between ephrin-B3 KO neurons and wild-type neurons in control MADM mice. In ephrin-B3 MADM mice, wild-type neurons had significantly higher spine density than heterozygous neurons, while heterozygous neurons were not significantly different than ephrin-B3 knockout neurons (Figure 6C). This suggests that in terms of their ability to compete for synapses, ephrin-B3 heterozygous neurons may be closer to ephrin-B3 knockout neurons than wild-type neurons. If this is the case, the competitive advantage of wild-type neurons in the context of the ephrin-B3 heterozygous background would be greater than the disadvantage of the ephrin-B3 knockout neurons. This could explain why spine density on wild-type neurons within a heterozygous background is higher than in neurons in control MADM mice, but spine density on ephrin-B3 knockout neurons within a heterozygous background is not significantly lower than that of neurons in control MADM mice. This is now added to the eighth paragraph of the Discussion section.